**Data Availability Statement:** All relevant data are within the paper and its Supporting Information files.

# Correlation among experience of person-centered maternity care, provision of care and women's satisfaction: Cross sectional study in Colombo, Sri Lanka

**Mohamed Rishard**[1,2]*, **Fathima Fahila Fahmy**[2], **Hemantha Senanayake**[1,2], **Augustus Keshala Probhodana Ranaweera**[1], **Benedetta Armocida**[3], **Ilaria Mariani**[3], **Marzia Lazzerini**[3]

**1** University Obstetrics Unit, De Soysa Hospital for Women, Colombo, Sri Lanka, **2** Department of Obstetrics & Gynaecology, Faculty of Medicine, University of Colombo, Colombo, Sri Lanka, **3** Institute for Maternal and Child Health—IRCCS "Burlo Garofolo"—Trieste, Italy

* rishi7875@yahoo.com

## Abstract

Person-centered maternity care (PCMC) is defined as care which is respectful of and responsive to women's and families' preferences, needs, and values. In this cross-sectional study we aimed to evaluate the correlations among the degree of PCMC implementation, key indicators of provision of care, and women's satisfaction with maternity care in Sri Lanka. Degree of PCMC implementation was assessed using a validated questionnaire. Provision of good key practices was measured with the World Health Organization (WHO) Bologna Score, whose items include: 1) companionship in childbirth; 2) use of partogram; 3) absence of labor stimulation; 4) childbirth in non-supine position; 5) skin-to-skin contact. Women's overall satisfaction was assessed on a 1–10 Likert scale. Among 400 women giving birth vaginally, 207 (51.8%) had at least one clinical risk factor and 52 (13.0%) at least one complication. The PCMC implementation mean score was 42.3 (95%CI 41.3–43.4), out of a maximum score of 90. Overall, while 367 (91.8%) women were monitored with a partogram, and 293 (73.3%) delivered non-supine, only 19 (4.8%) did not receive labour stimulation, only 38 (9.5%) had a companion at childbirth, and 165 (41.3%) had skin-to-skin contact immediately after birth. The median total satisfaction score was 7 (IQR 5–9). PCMC implementation had a moderate correlation with women's satisfaction (r = 0.58), while Bologna score had a very low correlation both with satisfaction (r = 0.12), and PCMC (r = 0.20). Factors significantly associated with higher PCMC score were number of pregnancies (p = 0.015), ethnicity (p<0.001), presence of a companion at childbirth (p = 0.037); absence of labor stimulation (p = 0.019); delivery in non-supine position (p = 0.016); and skin-to-skin contact (p = 0.005). Study findings indicate evidence of poor-quality care across several domains of mistreatment in childbirth in Sri Lanka. In addition, patient satisfaction as an indicator of quality care is inadequate to inform health systems reform.

**Funding:** University grants commission funding for higher studies, 2017.

**Competing interests:** None competing interest.

**Abbreviations:** ANOVA, Analysis of Variance; CS, Caesarean section; IQR, Interquartile range; IUGR, Intrauterine growth restriction at ultrasound; LMIC, Low-middle income countries; MMR, Maternal mortality rate; PCMC, Person-centered maternity care; SDG, Sustainable Developmental Goals; SOP, Standards operating procedures; STROBE, Strengthening the Reporting of Observational Studies in Epidemiology; WHO, World Health Organization.

# Introduction

According to the most recent national Maternal Death Surveillance and Response system estimates, in 2017 the maternal mortality rate (MMR) in Sri Lanka was 33.8/100.000, over 99% of women received antenatal care, and 99.5% of births were attended by skilled health personnel [1, 2]. Although Sri Lanka is classified as a lower middle-income country, it has achieved a major decline in maternal mortality rates over the last sixty years—from 1694/100,000 in 1947 —reaching one of the lowest rates in the South Asian Region [1]. These remarkable gains have been obtained through consistent government commitment to health and specific health-related policies, including as critical aspects the provision of education and health services free of charge [3, 4].

Despite these achievements, there have been criticisms and concerns about the quality of maternal health care in Sri Lanka [1, 5–9]. In fact, although the Sustainable Developmental Goals (SDG) for Sri Lanka aim to reduce MMR to 25 (per 100,000 live births) by 2020 and to less than 10 (per 100 000 live births) by 2030 [10], MMR has remained static at 31 to 39 deaths per 100,000 live births for almost a decade [1, 2, 6]. In 2017, according to national reports, nearly 70% of the maternal deaths were categorized as preventable [1, 2], and sub-optimal care both at community and hospital levels contributed to 38% of maternal deaths [1, 2]. Evident gaps are reported in service delivery, such as non-adherence to clinical protocols and standard practices [1, 2, 8]. Additionally, inappropriate practices have been described, such as the rising rate of caesarean section (CS), reaching nearly 45% in selected facilities [9], and the increasing rate of induction of labour, the highest in Asia (35.5%), and estimated to being performed without a medical indication in about 27.8% of cases [11].

Other aspects of quality of care deserve additional attention. The World Health Organization (WHO) Quality of Care Framework for maternal and newborn health [12] highlights the importance of considering the "experience of care" as a critical dimension, together with the "provision of care", which should include evidenced-based practices. Key aspects of the "experience of care" include effective communication, respect and dignity, and emotional support [12]. The importance of person-centered maternity care (PCMC), which is care respectful of and responsive to women's and families' preferences, needs, and values [13, 14], also defined patient-centred, people-centred, or woman-centred care, has been further emphasized in WHO recommendations for a positive childbirth experience [15]. There is evidence that PCMC has not been given enough attention in Sri Lanka's maternal care system [16–20]. Indeed, despite explicit WHO recommendations for labor companionship as a low-cost intervention to improve labor outcomes [16] and its inclusion in Sri Lankan national policy [17], a recent survey highlighted that nearly 60% of consultant obstetricians did not allow labour companions in their wards [18]. Although few studies have explored the area of mistreatment and abuse of women during pregnancy in south Asia, existing qualitative reports suggest a tendency for discriminatory behavior (such as verbal, emotional and even sexual abuse) and a diffuse normalization of disrespectful and abusive treatment of female patients [19, 21]. To author's knowledge, no quantitative study has yet been conducted on women's perspectives of PCMC in Sri Lanka. There are also no reports analyzing the correlation between indicators of PCMC and indicators of "provision of care" and women's overall satisfaction with maternal care. This study aimed to explore different domains of quality of care–namely degree of implementation of PCMC, key indicators of provision of care, and women's satisfaction with maternal care–in a tertiary care center in Sri Lanka, and to analyse correlations among these three domains, as well as key factors associated with each domain.

## Methods

### Study design and setting

This was a cross-sectional study, reported using the standards for Strengthening the Reporting of Observational Studies in Epidemiology (STROBE) [22]. The STROBE checklist for cross-sectional studies is provided in S1 Table.

The study was conducted in the Labor and Maternity wards of the University Obstetrics Unit of De Soysa Hospital for Women, the largest Maternity Unit in Sri Lanka, from December 2018 to April 2019.

### Study population

During the study period all women who delivered vaginally (including operative vaginal births), aged 15 to 45 years were considered for inclusion. Women who underwent a CS, were outside the indicated age range, diagnosed with major psychiatric illnesses, hospitalized in an intensive care unit, or refused consent were excluded. Eligible mothers were identified using the unit birth registry. All consecutive deliveries were screened for inclusion criteria. Eligible women who consented to participate in the study were interviewed. Deliveries that took place during day-time, nights and weekends were included.

### Data collection procedures

**Women's characteristics and health outcomes.** Women's characteristics and health outcomes were collected prospectively through an individual-patient database, established as method of routine data collection at De Soysa hospital since 2015 [23]. Detailed methods of data collection for this database have been previously reported [23, 24]. Briefly, maternal socio-demographic characteristics, medical risk factors, process indicators, and maternal and neonatal health outcomes were collected for each individual birth using a standardised two-page form and entered in real time in an electronic database by trained staff. Data quality assurance procedures included use of detailed case definitions, standard operating procedures, regular random checks, and 137 automatic validation rules aimed at minimizing data entry errors, and resulting, as previously reported, in high quality data [23].

**Person-centered maternity care.** Patient-centered maternity care was assessed using the PCMC questionnaire, which has been validated by Afulani et. al. in similar settings (India, Kenya) and which shows high content, construct, and criterion validity and good internal consistency reliability, as described in detail elsewhere [14, 25]. The questionnaire includes 30 items on three key domains: 1) dignity and respect, 2) communication and autonomy, and 3) supportive care. Each item has a four-point response scale, 0 ("no, never"), 1 ("yes, a few times"), 2 ("yes, most of the time"), and 3 ("yes, all the time"). The total score can therefore range from 0 to 90, with higher scores representing better care (**S2 Table**). The full scale and subscales have good internal-consistency reliability, with a Cronbach's α value of over 0.8 for the full scale across all groups and ranging between 0.61 and 0.75 for the subscales [26].

Before starting the study, the questionnaire was translated into the local languages (Tamil and Sinhalese) and back-translated to ensure consistency with the original version. The questionnaire was administered in the immediate post-natal period, before discharge, by an independent trained female researcher. All interviews were conducted in Sinhala or Tamil following pre-defined standard operating procedures. Both the interviewers' outfit/uniform and her identification card clearly identified her as a non-staff member. The interviewer introduced herself, the objective of the interview, and clarified that the interviews was anonymous. The interview was conducted in a separate area with appropriate privacy.

**Provision of care.** In order to measure key "good practices", we used the Bologna score, a simple score developed by the WHO and widely used for surveys [27–29]. The score assesses the following five key "good-practices": 1) presence of a companion at the time of birth; 2) use of partograph; 3) absence of labor stimulation (use of oxytocin, external pressure of the uterine fundus, or episiotomy); 4) delivery in non-supine position; 5) skin-to-skin contact with their newborn immediately post-partum. Of these five measures, four are considered as "provision of care" by the WHO framework (12), while companionship in labour has been debated for long time, since evidence show direct benefit on health outcomes. Each measure is assigned a score: "1" if present and "0" if missing. The total score is calculated as the sum of the score of all measures. Information on these indicators were extracted directly from the patients' files and verified, where appropriate, with women during the interview.

**Women's satisfaction.** Total satisfaction with care received was measured using a Likert scale from 1 (min) to 10 (max). The information was collected from mothers before discharge by a trained independent researcher, following the same procedures specified for the PCMC questionnaire.

## Sample size

A sample of 385 women was estimated as needed to detect, at a 95% confidence level and 5% margin of error, a normalized to 100 PCMC scale of 50, as expected based on existing literature [26]. An additional 15 women were recruited to ensure statistical significance in the case of missing data.

## Data analysis

Statistical analysis included descriptive statistics, correlation tests, and univariate and multi-variate analysis to examine associations between dependent and independent variables.

First, we calculated descriptive statistics with absolute frequencies and percentages for categorical variables. Continuous variables, i.e. PCMC scale, Bologna score and total satisfaction, were tested for normal distribution with the Shapiro-Wilk test. Variables normally distributed were reported as means and 95% confidence intervals (95%CI), while non-normally distributed variables were reported as median and interquartile range (IQR). To allow easy comparison across the different PCMC domains, re-scaled scores were calculated as the fraction of the total possible score on each domain and normalised to 100.

Next, correlations among scores were evaluated with Pearson's correlation test for normally distributed variables and with Spearman's rank correlation test for non-normally distributed variables.

Thirdly, we conducted a t-test or a one-way analysis of variance (ANOVA) to examine mean differences in normally distributed scores (i.e., PCMC and Bologna score) among different categories of maternal characteristics (age, number of pregnancies, education, occupational status, ethnicity, presence of clinical risk factors), process indicators (labour onset, mode and hour of delivery), adverse health outcome (defined as occurrence of one of the following maternal complications: sepsis or severe infection, postpartum hemorrhage, III-IV degree perineal tears, or near miss) or single components of the Bologna score. Significant variables in the bivariate analysis were included in a multi-variate ANOVA. Bonferroni correction was used for multiple comparisons. For scores which deviate from the normality assumption, i.e. women's satisfaction, a logistic regression model was fit and odds ratios (OR) were reported for each predictor. Factors resulting as significantly associated in bivariate analysis were included in a multivariate logistic regression. To perform this analysis women's

satisfaction was dichotomized at the minimum satisfaction limit of 6 (Likert scale equal or more than 6 versus Likert scale less than 6).

Lastly, a sensitivity analysis was performed dichotomizing women's satisfaction for the logistic regression model at the median value of the satisfaction Likert scale in case this value differed from the minimum satisfaction limit of 6.

A p value of less than 0.05 was considered significant. Statistical analysis was performed with STATA 14.0 (Stata Corporation, College Station TX) and SAS (Statistical Analysis Software 9.4 Institute Srl, Milan, Italy).

### Patient and public involvement

Women were involved in the study by providing their views on the quality of care received. Additionally, the development of data collection tools was informed by patients' experiences, as reported in literature [14, 23, 24, 27]. Users of maternity services at De Soysa Hospital will be involved in the next phases of the project to identify and agreed upon actions to improve quality of care around the time of childbirth.

### Ethical considerations

The study was approved by the Ethics Review Committee of the Faculty of Medicine, University of Colombo (Reference number EC-18-128). Before conducting the interviews, permission was obtained from the director of De Soysa Hospital for women. Written informed consent forms were provided in Tamil, English and Singhala and signatures were attained after having provided information about the research. In case of minors, informed assent and consent from guardians / parents were collected. Confidentiality was maintained by de-identifying all files before database entry.

## Results

### Women's characteristics

Among the 400 included women (**Table 1**), the majority (87.3%) were between 19–34 years old. Nearly half (45.3%) were primigravidae, while about a quarter (27.2%) had one previous pregnancy, and another quarter (27.5%) two previous pregnancies. Nearly half (45.0%) of women were Sinhalese, about one third (34.5%) Muslim, and 10.3% Tamil. Almost all women (99.7%) were married, while 83.5% were unemployed, and 90.7% had a secondary education. Overall, 207 (51.8%) women presented with at least one medical risk factor, with the most prevalent being gestational diabetes (21.3%). In about one third (32.7%) of women labor was induced, while few (3.0%) had an operative vaginal delivery. Nearly half (45.8%) of women delivered during night hours, two thirds (61.7%) were assisted by a nurse, one third (33.7%) by a midwife, and only a percentage by a doctor. Overall, 52 (13.0%) women had at least one adverse health outcome, with the most prevalent being near-miss cases (8.3%), post-partum hemorrhage (4.8%), II-IV degree perineal tears (6.8%), and sepsis (1.2%).

### Person-centered maternity care

PCMC scores were normally distributed (Shapiro-Wilk p = 0.299) as shown in **Fig 1A and 1B**.

Re-scaled PCMC scores (normalised to 100) are showed in Fig 2. The rescaled PCMC mean score for "communication and autonomy" was significantly lower (34.6; 95%CI 33.2–36.0) compared to the score of the other domains (full PCMC score: 47.1; 95%CI 45.9–48.2; "dignity and respect": 57.2; 95%CI 55.8–58.6; "supportive-care": 50.5; 95%CI 49.0–51.9; adjusted p≤0.002 for all comparisons). PCMC not rescaled values are reported in **S2 and S3 Tables**.

**Table 1. Women's characteristics.**

| | n (N = 400) | % |
|---|---|---|
| **Age** | | |
| < 18 years | 6 | 1.5 |
| 19–24 years | 133 | 33.3 |
| 25–34 years | 216 | 54.0 |
| 35–39 years | 40 | 10.0 |
| >40 years | 5 | 1.3 |
| **Number of pregnancies** | | |
| 1 | 181 | 45.3 |
| 2 | 109 | 27.2 |
| ≥3 | 110 | 27.5 |
| **Ethnic/religious group[1]** | | |
| Burger | 1 | 0.3 |
| Muslim | 138 | 34.5 |
| Sinhalese | 184 | 45.0 |
| Tamil | 77 | 10.3 |
| **Marital status** | | |
| Married | 399 | 99.7 |
| Unmarried | 1 | 0.3 |
| **Employed** | | |
| Yes | 66 | 16.5 |
| No | 334 | 83.5 |
| **Education** | | |
| None or Primary | 1 | 0.3 |
| Secondary | 363 | 90.7 |
| Higher | 36 | 9.0 |
| **Women with medical risk factors (any)[2]** | 207 | 51.8 |
| **Key maternal medical risk factors** | | |
| Gestational diabetes | 85 | 21.3 |
| Obesity | 48 | 12.0 |
| Gestational age at delivery >41 weeks | 32 | 8.0 |
| Gestational age at delivery <37 weeks | 25 | 6.3 |
| IUGR | 23 | 5.8 |
| Gestational hypertension | 13 | 3.3 |
| Maternal hypothyroidism | 10 | 2.6 |
| Previous CS | 10 | 2.6 |
| Others | 28 | 7.4 |
| **Labour onset** | | |
| Spontaneous | 269 | 67.3 |
| Induction | 131 | 32.7 |
| Pre-labour caesarean section | 0 | 0 |
| **Mode of delivery** | | |
| Vaginal spontaneous | 388 | 97.0 |
| Vaginal operative | 12 | 3.00 |
| **Episiotomy** | 358 | 89.5 |
| **Time of delivery** | | |
| Day (from 7 AM to 6 PM) | 214 | 53.5 |

(*Continued*)

**Table 1.** (Continued)

|  | n | % |
| --- | --- | --- |
|  | **(N = 400)** |  |
| Night (from 7 PM to 6 AM) | 183 | 45.8 |
| Missing[3] | 3 | 0.8 |
| **Health professional delivering care at birth** |  |  |
| Nurse | 247 | 61.7 |
| Midwife | 135 | 33.7 |
| House Officer | 0 | 0 |
| Senior house Officer | 5 | 1.3 |
| Registrar | 11 | 2.7 |
| Consultant | 1 | 0.3 |
| Missing | 1 | 0.3 |
| **Women with at least one adverse outcome** | 52 | 13.0 |
| **Key maternal adverse health outcomes** |  |  |
| Admission to intensive care unit | 0 | 0 |
| Near-miss cases [4] | 33 | 8.3 |
| Postpartum hemorrhage | 19 | 4.8 |
| Operating theatre after delivery | 0 | 0 |
| Hysterectomy | 0 | 0 |
| Uterine Rupture | 0 | 0 |
| Sepsis | 5 | 1.2 |
| Deep vein thrombosis/ Pulmonary embolism | 0 | 0 |
| Abruptio placentae | 0 | 0 |
| Amniotic fluid embolisms | 0 | 0 |
| Perineal tears III-IV degree | 24 | 6.0 |

[1] Muslim was included in this group since in Sri Lanka this is a well-defined community.

[2] Medical risk factors included in this category were: maternal age >40 years; gestational age <37 > = 41; obesity; multiple pregnancies; pre-gestational hypertension; gestational hypertension; pre-eclampsia; eclampsia; fetal malformation; chorioamnionitis; intrauterine growth restriction at ultrasound; gestational diabetes; pre-gestational diabetes; maternal cardiac disease; maternal hypothyroidism; polyhydramnios; oligohydramnios; antepartum hemorrhage; severe anaemia; previous caesarean section.

[3] Where missing cases were zero, they were not reported in the table.

[4] Near-miss cases were pre-defined based on locally agreed criteria as recommended by the WHO Manual" Evaluating the quality of care for severe pregnancy complications: the WHO near-miss approach for maternal health". Near miss cases were defined as: severe disease (severe PPH, severe pre-eclampsia, eclampsia, sepsis, uterine rupture, severe complications of abortion) OR critical events (admission UTI, intervention radiology, laparotomy, blood transfusion) OR organ dysfunction occurred during pregnancy, childbirth or within 42 days of termination of pregnancy.

Abbreviations: IUGR = intrauterine growth restriction at ultrasound; CS = caesarean section; APH = antepartum hemorrhage.

Frequencies of each of the 30 items on the PCMC scale is detailed in **S4 Table**. About two thirds of women (63.5%) reported that medical staff treated them with respect even though the majority (99.3%) reported to have been treated in an unfriendly manner. Overall, one out of six (14.8%) felt to have been treated roughly—pushed, beaten, slapped, pinched, physically restrained, or gagged—and nearly one third (28.5%) reported to have been shouted at, scolded, insulted, threatened, or talked to rudely. Most women (85.8%) reported that health professionals did not explain drugs that were administered, and more than half (55%) did not feel

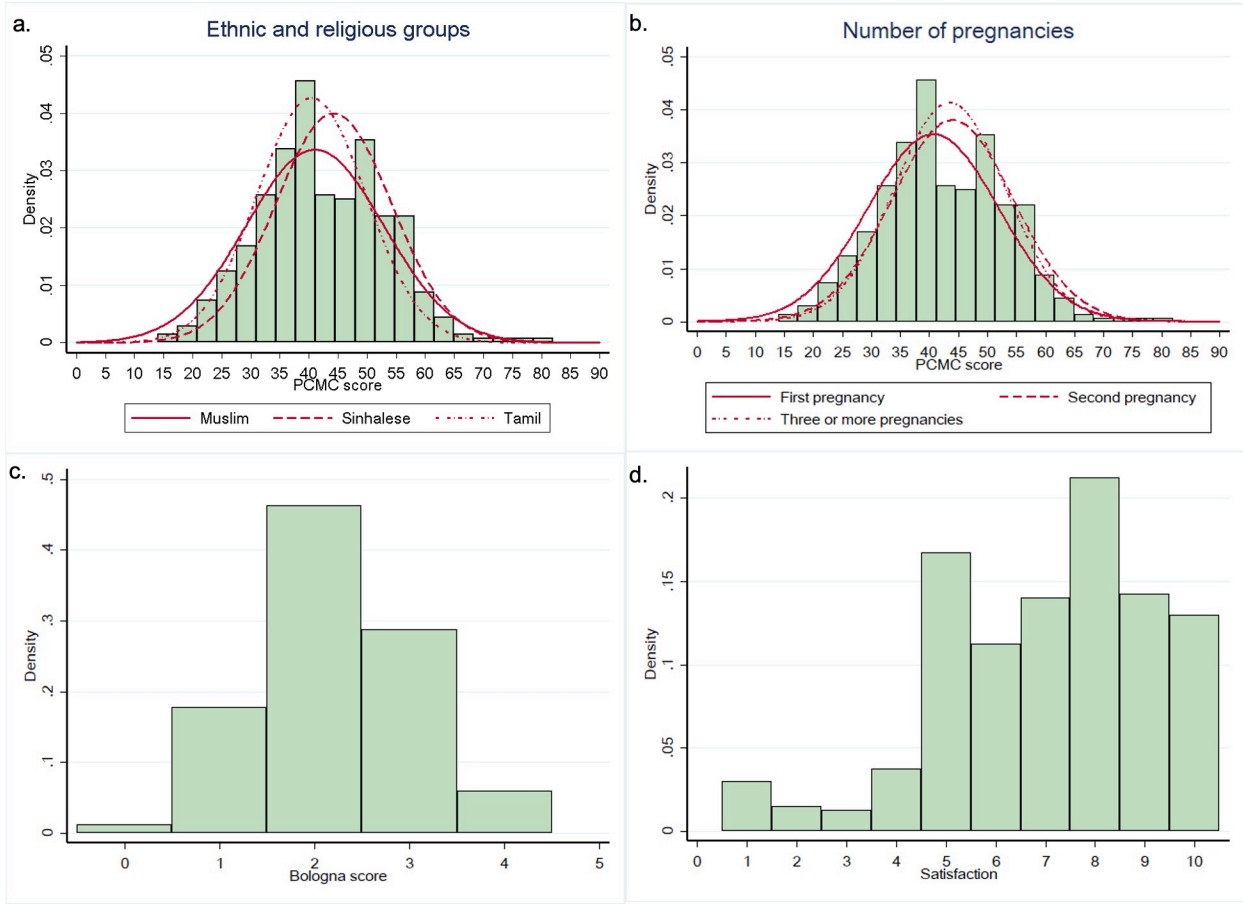

**Fig 1. Distribution of PCMC, Bologna score and women's satisfaction Likert scale.** Panel a) PCMC score distribution by ethnic and religious groups. Panel b) PCMC score distribution by number of pregnancies. Panel c) Bologna score distribution. Panel d) Women's satisfaction Likert scale distribution. PCMC score distribution by groups was added for significant variables in multivariate analysis different from Bologna score components.

involved in decisions about their care, nor were asked for permission or consent before performing procedures (57%). Less than a quarter (21.0%) thought that health professionals took the best of care of them or did everything they could to help control their pain (21.8%).

### Provision of care

The Bologna score was normally distributed (Shapiro-Wilk p = 0.994) (**Fig 1C**). Detailed findings on the Bologna Score are depicted in **Fig 3**. Out of all births, 367 (91.8%) were monitored with partogram, and 293 (73.3%) occurred in non-supine position. However, only 19 (4.8%) women did not have stimulation of labour, and only 165 (41.3%) had skin-to-skin contact with the baby immediately after delivery. Only 38 (9.5%) of 400 women in this sample had a companion at the time of birth. When this percentage was recalculated on the denominator of women who were not missing data on the reason for not having a birth companion, the recalculated percentage was 11.7%.

Reasons reported by women for the absence of a birth companion are further detailed in Table 2. The three main reason were: staff not allowing a birth companion (31.5% of all births);

**Fig 2. Re-scaled PCMC scores.**

practical problems which prevented the companion to be present (e.g., need to look after other children, or to work) (21.3%); women reporting not wishing to have a companion during childbirth (19%).

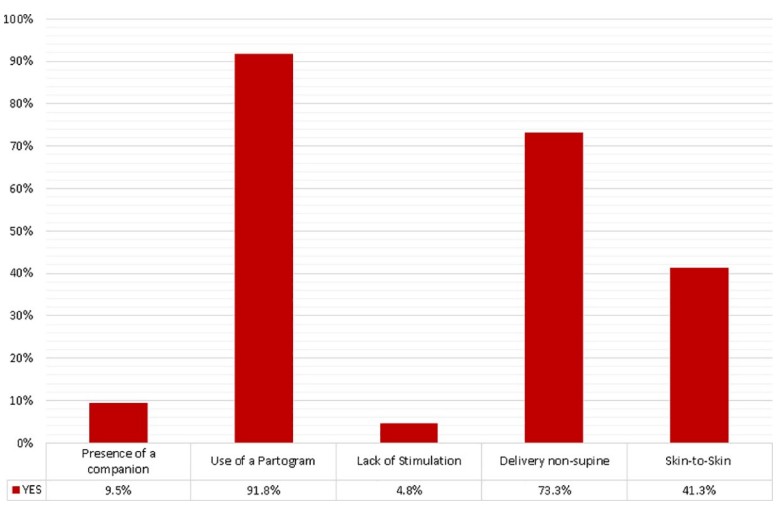

**Fig 3. Findings of the Bologna score questionnaire.**

**Table 2. Reasons for absence of a companion at the time of birth.**

| | n | % |
|---|---|---|
| | (N = 356) | |
| Staff did not allow the presence of a companion | 112 | 31.5 |
| Practical problems (e.g., older child who needed to be looked after, companion not being available due to employment, or being far away) | 76 | 21.3 |
| Woman reporting not wanting to have a companion | 76 | 21.3 |
| My labour was too quick / No time to inform | 34 | 9.6 |
| Presence of a companion was not possible due to companion's medical condition | 25 | 7.0 |
| I was not aware about this possibility | 19 | 5.3 |
| Other | 14 | 3.9 |

## Women's satisfaction

Women's overall satisfaction was not normally distributed (Shapiro-Wilk p<0.001) as shown in **Fig 1D**. The median satisfaction score was 7 (IQR range: 5 to 9) with 295 women (73.7%) above the minimum satisfaction limit of 6; 186 (46.5%) had a satisfaction score between 6–8, and 109 (27.3%) a satisfaction score of >8 out of 10.

## Correlation among scores

A low but significant correlation was observed (Pearson r = 0.20, p<0.001) between the PCMC scale and the Bologna score (**Fig 4A**), and between Bologna score and total satisfaction (Spearman r = 0.12, p = 0.018) (**Fig 4B**) while a moderate correlation was found between PCMC and total satisfaction (Spearman r = 0.58, p<0.001) (**Fig 4C**). Further details are provided in **S5** and **S7** **Tables**.

Comparing each PCMC sub-domain with the satisfaction score, the sub-domain most strongly correlated with satisfaction was supportive care (Spearman r = 0.55, p<0.001), which included 15 of the total 30 items of the full PCMC scale, while the other sub-scores had a low correlation (dignity and respect: Spearman r = 0.43, p<0.001; communication and autonomy: Spearman r = 0.35, p<0.001) (**S6 Table**).

## Univariate and multivariate analyses

Results of these analyses are reported in detail in **Table 3**. From the univariate analysis, the mean PCMC score was significantly higher in Sinhalese women compared to Muslim (mean difference: 3.3; p = 0.007) and to Tamil (mean difference: 3.8; p = 0.008) women. Similarly, women in their second pregnancy had a significantly higher mean PCMC score than women at their first pregnancy (mean difference: 3.3; p = 0.030). Distribution of PCMC score by ethnic and religious groups and by number of pregnancies is shown in Fig 1A and 1B, respectively. PCMC score was significantly higher in case of presence of a companion at childbirth (mean difference: 3.8; p = 0.037); absence of labor stimulation (mean difference: 5.9; p = 0.019); delivery in non-supine position (mean difference: 2.9; p = 0.016); and skin-to-skin contact (mean difference: 3.0; p = 0.005). Significance of these factors were confirmed in a multivariate model.

The mean Bologna score was significantly higher in women who delivered during day-time compared to night-time (mean difference: 2.2; p = 0.045). No significant differences were found in variables, except for the single components of the Bologna score, therefore no multivariate analysis was performed.

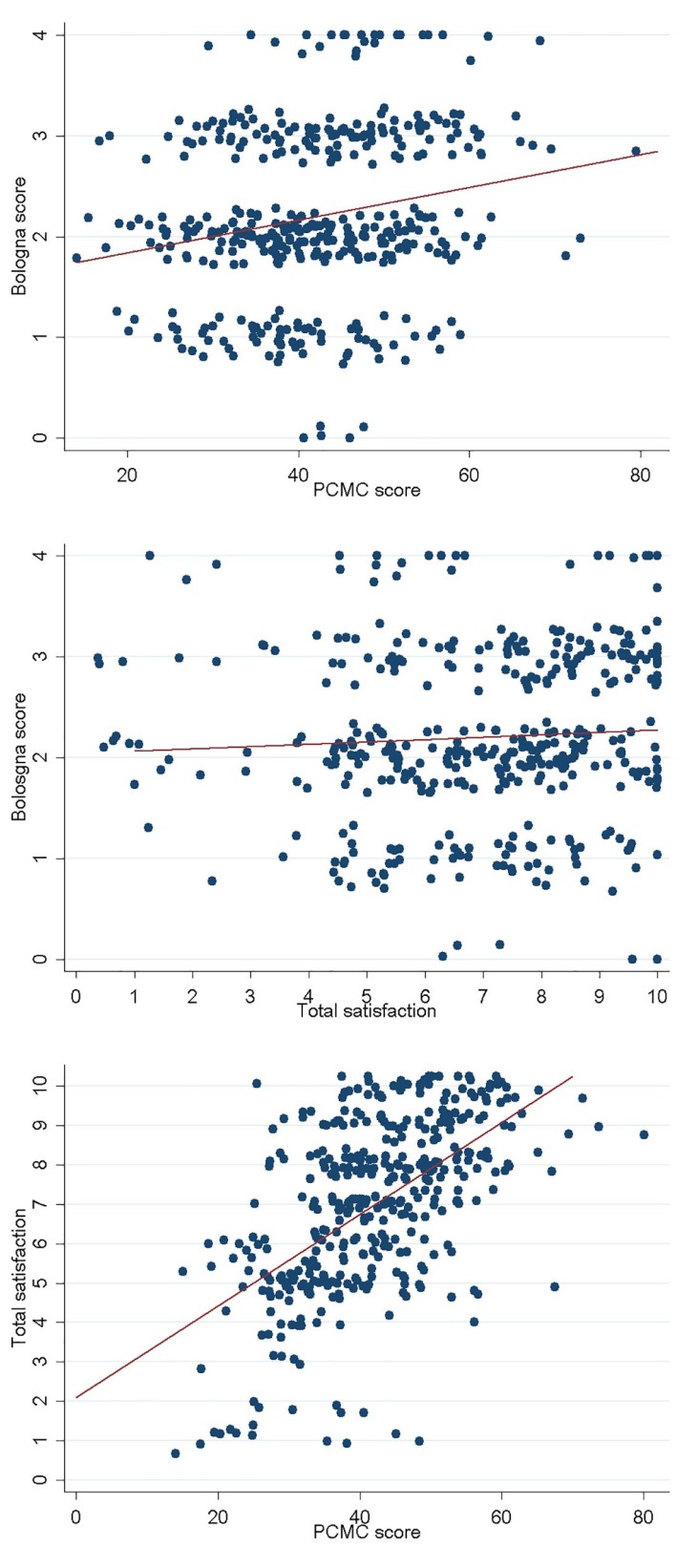

**Fig 4. Correlation among different indicators.**

**Table 3. Two- and multi-ways ANOVA.**

| | PCMC score | | | Bologna score | | Satisfaction | |
|---|---|---|---|---|---|---|---|
| | Unadjusted mean score | Bivariate analysis | Multivariate analysis* | Unadjusted mean score | Bivariate analysis | Bivariate analysis | |
| | | | | | | Crude OR | p value |
| | (95% CI) | (p value) | (p value) | (95% CI) | (p value) | (95% CI) | |
| **Age** | | | | | | | |
| < 18 years | 36.3 (31.3–41.3) | 0.122 | - | 2.2 (1.6–2.8) | 0.906 | 1.54 (0.18–13.53) | 0.694 |
| years | 41.0 (39.1–42.0) | 0.068 | - | 2.2 (2.1–2.3) | 0.955 | 0.67 (0.41–1.08) | 0.103 |
| years | 43.1 (41.8–44.6) | Ref | - | 2.2 (2.1–2.3) | Ref | Ref | Ref |
| 35–39 years | 42.7 (40.0–45.4) | 0.795 | - | 2.2 (1.9–2.5) | 0.955 | 0.81 (0.38–1.75) | 0.598 |
| >40 years | 44.2 (38.0–50.4) | 0.897 | - | 2.2 (1.8–2.6) | 0.983 | - | - |
| **Number of pregnancies** | | | | | | | |
| 1 | 40.6 (39.0–42.3) | Ref | Ref | 2.2 (2.1–2.3) | Ref | Ref | Ref |
| 2 | 44.0 (42.0–46.0) | 0.010 | 0.015 | 2.2 (2.1–2.4) | 0.713 | 1.39 (0.80–2.42) | 0.247 |
| ≥3 | 43.4 (41.7–45.3) | 0.026 | 0.017 | 2.2 (2.0–2.4) | 0.661 | 1.05 (0.62–1.78) | 0.867 |
| **Education** | | | | | | | |
| None or Primary | 42.0 (42.0–42.0) | 0.996 | - | 2.0 (2.0–2.0) | 0.815 | - | - |
| Secondary | 42.0 (40.9–43.2) | Ref | - | 2.2 (2.1–2.3) | Ref | Ref | Ref |
| Higher | 45.3 (41.9–48.8) | 0.080 | - | 2.3 (1.9–2.5) | 0.592 | 1.88 (0.76–4.64) | 0.174 |
| **Employed** | | | | | | | |
| No | 42.2 (41.1–43.4) | Ref | - | 2.2 (2.1–2.3) | Ref | Ref | Ref |
| Yes | 42.9 (39.9–45.9) | 0.660 | - | 2.3 (2.1–2.5) | 0.303 | 1.14 (0.61–2.10) | 0.685 |
| **Ethnic/religious group** | | | | | | | |
| Muslim | 41.0 (39.0–43.0) | Ref | Ref | 2.2 (2.1–2.4) | Ref | Ref | Ref |
| Sinhalese | 44.3 (42.8–45.7) | 0.007 | 0.002 | 2.2 (2.1–2.3) | 0.805 | 1.42 (87–2.36) | 0.163 |
| Tamil | 40.5 (38.4–42.5) | 0.702 | 0.750 | 2.2 (1.9–2.4) | 0.723 | 1.13 (0.61–2.10) | 0.705 |
| Burger | 18.0 (18.0–18.0) | 0.030 | 0.049 | 2.0 (2.0–2.0) | 0.792 | - | - |
| **Risk factors (any)** | | | | | | | |
| No | 42.6 (41.0–44.1) | Ref | - | 2.3 (2.1–2.4) | Ref | Ref | Ref |
| Yes | 42. (40.7–43.6) | 0.724 | - | 2.1 (2.0–2.3) | 0.176 | 0.83 (0.53–1.29) | 0.405 |
| **Labour onset** | | | | | | | |
| Spontaneous | 41.7 (40.4–42.9) | Ref | - | 2.2 (2.1–2.3) | Ref | Ref | Ref |
| Induction | 43.7 (41.9–45.6) | 0.062 | - | 2.2 (2.1–2.4) | 0.517 | 1.39 (0.84–2.26) | 0.193 |
| **Mode of delivery** | | | | | | | |
| Vaginal spontaneous | 42.3 (41.2–43.4) | Ref | - | 2.2 (2.1–2.3) | Ref | Ref | Ref |
| Vaginal operative | 43.3 (38.5–48.2) | 0.747 | - | 1.8 (1.4–2.2) | 0.122 | 4.03 (0.51–31.59) | 0.185 |
| **Hour of delivery** | | | | | | | |
| Night (from 7 PM to 6 AM) | 41.5 (40.0–43.0) | Ref | - | 2.1 (1.9–2.2) | Ref | Ref | Ref |
| Day (from 7 AM to 6 PM) | 42.8 (41.3–44.2) | 0.214 | - | 2.2 (2.1–2.4) | 0.045 | 0.87 (0.56–1.37) | 0.553 |
| **Adverse health outcomes** | | | | | | | |
| No | 42.6 (41.5–43.8) | Ref | - | 2.2 (2.1–2.3) | Ref | Ref | Ref |
| Yes | 40.4 (37.1–43.6) | 0.153 | - | 2.0 (1.8–2.6) | 0.178 | 0.63 (0.34–1.17) | 0.144 |
| **Bologna score components** | | | | | | | |
| Presence of a companion | | | | | | | |
| No | 42.0 (41.0–45.8) | Ref | Ref | 2.1 (2.0–2.2) | Ref | Ref | Ref |
| Yes | 45.8 (41.9–49.68) | 0.037 | 0.023 | 3.3 (3.1–3.6) | < .001 | 0.51 (0.25–1.01) | 0.055 |
| Use of partograph | | | | | | | |

*(Continued)*

**Table 3.** (Continued)

| | PCMC score | | | Bologna score | | Satisfaction | |
|---|---|---|---|---|---|---|---|
| | Unadjusted mean score | Bivariate analysis | Multivariate analysis* | Unadjusted mean score | Bivariate analysis | Bivariate analysis | |
| | | | | | | Crude OR | p value |
| | (95% CI) | (p value) | (p value) | (95% CI) | (p value) | (95% CI) | |
| No | 42.9 (39.1–46.6) | Ref | - | 1.2 (0.9–1.5) | Ref | Ref | Ref |
| Yes | 42.3 (41.2–43.4) | 0.780 | - | 2.3 (2.2–2.4) | < .001 | 0.60 (0.24–1.50) | 0.276 |
| Absence of stimulation to labor | | | | | | | |
| No | 42.1 (40.99–43.2) | Ref | Ref | 2.2 (2.1–2.2) | Ref | Ref | Ref |
| Yes | 48.0 (43.4–52.5) | 0.020 | 0.026 | 3.3 (2.9–3.6) | < .001 | 0.99 (0.35–2.84) | 0.995 |
| Delivery in non-supine position | | | | | | | |
| No | 40.21 (38.2–42.2) | Ref | Ref | 1.4 (1.3–1.5) | Ref | Ref | Ref |
| Yes | 43.1 (41.9–44.4) | 0.016 | 0.017 | 2.5 (4.4–2.6) | < .001 | 1.29 (0.79–2.10) | 0.316 |
| Skin-to-skin care | | | | | | | |
| No | 41.10 (39.75–42.45) | Ref | Ref | 1.8 (1.7–1.8) | Ref | Ref | Ref |
| Yes | 44.1 (42.5–45.8) | 0.005 | 0.027 | 2.8 (2.7–2.9) | < .001 | 1.19 (0.76–1.89) | 0.445 |

* Significant variables in the bivariate analysis were included in a multivariate analysis.

Abbreviation: PCMC = Person-centered maternity care.

No factor was associated with women's satisfaction. However, in sensitivity analysis, number of pregnancy (OR of secondigravidae compared to primigravidae is 1.95, 95%CI 1.17–3.24, p = 0.010) and a delivery in non-supine position (OR 1.02, 95% CI 1.02–2.52, p = 0.039) were significantly associated with a satisfaction score above the median value of 7. Further details of frequencies and sensitivity analysis are shown in **S7 and S8 Tables**.

## Discussion

This is the first quantitative study, to our knowledge, conducted in Sri Lanka reporting data on women's views of PCMC and on the Bologna score. Additionally, this is one of the few studies in South East Asia reporting on PCMC [26], and possibly the first one exploring correlations between degree of implementation of PCMC, provision of key aspects of maternal care, and women's satisfaction. Study findings indicate evidence of poor-quality care across several domains of mistreatment in childbirth in Sri Lanka. Findings of the study suggest that all domains of PCMC and several aspects of provision of care require improvement in the study setting and suggest that further studies are needed to better document quality of maternal care across Sri Lanka. This study contributes to the growing body of evidence suggesting that "experience of care" is a key aspect of quality of care that warrants further attention [29–33]. Disrespectful and abusive behavior during childbirth and maternity care remain a global health problem [34–39], and there is still a lack of information and underestimation of the problem [40].

Results of this study also strongly suggest that patient satisfaction as an indicator of quality care is inadequate to inform health systems reform. The inconsistent distribution among different scores (PCMC, Bologna and satisfaction) and the poor correlation among them, suggest that satisfaction with care taken alone, as frequently done in hospital surveys [41, 42], is not be a good proxy for other domains of quality of care. Quality of care should be investigated using several other indicators, and should include an evaluation of mistreatment and PCMC as standalone indicators of quality, safety and rights.

Interestingly, women's satisfaction had a very poor correlation with the Bologna score, and a moderate correlation with PCMC. This suggests that women's satisfaction may have been more affected by the PCMC implementation than by provision of key aspects of maternal care, and that the two domains were weakly associated in women's minds. Other studies in Asia have observed a good correlation among key aspects of "experience of care", such as efficient communication, and participation to care, and overall patient satisfaction [42]. These findings call for further research to explore which factors are more strongly associated with women's overall satisfaction in different settings.

On the other hand, high reported women's satisfaction should be interpreted with caution, since it may be attributed to different factors, such as personal beliefs and values, ethnicity, religion, and the location of the facility [43, 44]. Studies have described that women tend to normalize disrespectful care when they experience good health outcomes [45, 46].

When compared to the few existing studies on PCMC, a previous survey conducted in rural Ghana [26] pointed out similar PCMC scores as observed in our study (mean PCMC score 46.5 SD 6.9), while interestingly, an evaluation in over 2000 women across 40 facilities in a rural setting in Uttar Pradesh, resulted in a higher score (mean PCMC score 55.8, SD 11.6) [26]. The higher score observed in India [26] compared to Sri Lanka may be explained by different factors, including differences in the quality of care received, in the population characteristics and in the setting. In a multivariate analysis conducted in the Indian sample, educated, employed and wealthier women reported a higher PCMC score than did uneducated unemployed and poorer women. In our sample only 0.3% of women had none or only primary education, compared to 47% in the Indian sample, and may therefore have been more empowered to express an opinion on PCMC. Similarly, a survey in four countries in Africa and Asia using another tool to explore experience of care reported that about one third of women were mistreated, and that frequency of mistreatment was higher in the younger and poorly educated [30], whilst a cross sectional study in Iran showed that three out of every four women reported perceived disrespectful maternity care [47]. Moreover, in India a mixed method study reported a total mistreatment scores higher amongst women attending district hospitals, women above 35 years of age, primiparous, and women belonging to the "scheduled caste and tribe" [48].

Our findings of different ethnic/religious groups reporting different PCMC scores may suggest a different "perception "or otherwise discrimination of these minority groups, as also described in other studies [20, 48, 49]. Better PCMC score with increasing number of previous pregnancies can be explained by their experience with health care system [20], and the fact that women tend to normalize the poor care with experience [50].

More studies should further explore women's views on PCMC in different settings—including high-income countries, where the few existing studies suggest that "experience of care" may still be unsatisfactory [30, 51]—and better document how education, ethnicity, social class empowerment and values affect the scores of the PCMC scale. The observation that higher PCMC scores are associated with lower education and expectations is clearly important to interpret and compare results across different settings.

Additionally, it will be interesting to further explore providers' perception of PCMC. Studies have found that incongruence between women's and providers' perceptions may negatively impact women's compliance, satisfaction, and use of health services [52]. In a recent study in Kenya women reported lower levels of PCMC compared to providers [52], while a study in Italy found that providers more frequently than mothers judged implementation of key items of PCMC as "inadequate", such as effective communication [51]. Furthermore, a recent qualitative study in Ghana conducted with midwives revealed that provider perception and victim blaming–with socio-economic inequalities and health system related factors—facilitated disrespect and abusive care [53].

Two recent studies used the Bologna score in countries in Asia: a study in Nepal found a mean Bologna score of 1.43 [54], while a study in Cambodia [29] observed higher Bologna scores. Case definitions were slightly different, e.g. presence of a companion was not strictly measured as presence during childbirth, but rather during labour and in the post-natal ward. Notably, in this study in Sri Lanka some of items of the Bologna score actually indicated good practices: for example, delivery in non-supine position was much more frequent in this study than what was reported in a study in Italy [51]. This difference may be related to over-use of cardiotocography during delivery in Italy for documenting fetal heart beat rate, a practice widely used for protection in case of legal disputes [51]. Lastly, some findings of the Bologna score deserve further evaluation. For example, considering that the use of partogram is mandatory at De Soysa hospital, it will be interesting to evaluate whether poor staffing and/or expedited deliveries can explain the observed frequency of use (92%).

Very little is known from previous literature on how the different scores of experience and provision of care are associated with each other and with health outcomes [55]. In a recent study conducted in Kenya, higher PCMC scores were significantly associated with willingness to return to the facility for the next delivery, a measure frequently used, together with other measures of satisfaction, to assess overall satisfaction with care received by women during childbirth. Moreover, this was associated with better newborn health outcomes [38]. Interestingly, in our study, none of the indicators evaluated, neither on experience of care, provision, nor overall satisfaction, was associated with the maternal health outcomes. On the other hand, the PCMC score differed significantly between various ethnic and religious groups, in women with a higher number of pregnancies, and the Bologna score between day-time and night-time. This should be further evaluated in other studies with a larger sample.

In terms of lessons for policy makers, this study indicates a need for action to ensure that every woman has access to the highest attainable standard of health, which includes the right to dignified, respectful healthcare [56]. The detailed findings of the PCMC scale—such as the frequency of mistreatment of women, lack of information and women's participation in care—should be used to develop interventions to promote PCMC. Efforts to improve PCMC may include provider training on the importance of PCMC, patients' and providers' rights, and strategies to improve providers' interactions with women and their families. Similarly, findings on the Bologna Score can be used to promote partnership in labor, and skin-to-skin contact between mother and baby, together with judicious use of oxytocin and restricted use of episiotomy. Data from this study may be used as a baseline against which to compare future post-intervention surveys.

Further research is needed to examine how to routinely collect woman-reported experiences of care, triangulate them with other data on provision of care and health outcomes, and how to use all of this integrated information to prioritize interventions to improve quality of care [57, 58]. This pilot experience can be of interest to both researchers and policymakers, as a relatively simple model to investigate different dimensions of quality of care.

We acknowledge as limitations the conduct of the study in a single center and the exclusion of women with psychiatric illnesses, which may be more likely than the general population of women to experience mistreatment. Further studies using tools specifically developed and validated for this population are needed to evaluate PCMC in women with mental illness.

## Supporting information

**S1 Table. STROBE statement—checklist of items that should be included in reports of cross-sectional studies.**
(DOCX)

**S2 Table. PCMC scale and sub-scales.**
(DOCX)

**S3 Table. Study findings on the PCMC questionnaire.**
(DOCX)

**S4 Table. Frequency of each item on the PCMC scale.**
(DOCX)

**S5 Table. Pearson correlation between the PCMC sub-scales and Bologna score.**
(DOCX)

**S6 Table. Spearman correlation between the PCMC sub-scales, Bologna score and total satisfaction.**
(DOCX)

**S7 Table. Absolute frequency and percentage of satisfaction score dichotomised at the minimum satisfaction limit of 6.**
(DOCX)

**S8 Table. Sensitivity analysis.** Absolute frequency and percentage of satisfaction score dichotomized at the median value of 7.
(DOCX)

**S1 File.**
(XLS)

## Acknowledgments

We would like to thank all women who participated in the study and the staff of the University Obstetric Unit of the De Soysa Hospital for Women.

## Author Contributions

**Conceptualization:** Mohamed Rishard, Hemantha Senanayake, Marzia Lazzerini.

**Data curation:** Fathima Fahila Fahmy.

**Formal analysis:** Ilaria Mariani.

**Resources:** Augustus Keshala Probhodana Ranaweera.

**Supervision:** Mohamed Rishard, Marzia Lazzerini.

**Writing – original draft:** Mohamed Rishard, Benedetta Armocida, Ilaria Mariani, Marzia Lazzerini.

**Writing – review & editing:** Mohamed Rishard, Benedetta Armocida, Ilaria Mariani, Marzia Lazzerini.

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
