## [Decision Letter · Decision Letter 0]

14 Sep 2020

PONE-D-20-18688

Correlation among experience of person-centered maternity care, provision of care and women’s satisfaction: cross sectional study in Colombo, Sri Lanka

PLOS ONE

Dear Dr. Rishard,

Thank you for submitting your manuscript to PLOS ONE. After careful consideration, we feel that it has merit but does not fully meet PLOS ONE’s publication criteria as it currently stands. Therefore, we invite you to submit a revised version of the manuscript that addresses the points raised during the review process.

We look forward to receiving your revised manuscript.

Kind regards,

Emma Sacks

Academic Editor

PLOS ONE

Journal Requirements:

Additional Editor Comments (if provided):

BEFORE SUBMITTING A REVISION, PLEASE ADD PAGE AND LINE NUMBERS. IT IS VERY DIFFICULT FOR REVIEWERS TO GIVE COMMENTS WITHOUT THESE.

While this is a critically important topic, and clearly understudied in Sri Lanka, this paper needs significantly more analytic work. The analyses are very basic and present only the totals, despite the methods indicating the ability to assess various correlations between demographics and practices, and between the PCMC, Bologna, and Likert scales. The results presented here are cursory. Detailed results should be in the paper, not supplementary appendices.

This paper requires more editing for grammar (even in the first paragraph, there are mistakes) and spelling (including Caesarean etc).

Please spell out acronyms, including PCMC, IUGR etc, at first use and in tables.

Please include in limitations that not including women with psychiatric illnesses may introduce bias, as these women may be more likely to experience mistreatment.

Please include information about how women below age 18 were consented/assented.

Please include more specifics in the methods - how were women recruited? what is an "established individual patient database" = are these hospital records?

Please include more details about the sample size calculation.

Please explain how it was ensured that women understood that research assistants were not hospital staff (which may have led to desirability bias)

Why were indicators chosen like "non supine" rather than "women's preferred birthing positions were respected"? Wouldn't that be more person-centered?

When referring to "risk factors" please clarify "medical risk factors" (to differentiate from social risk)

"Near miss" is vague for a medical event - can you include the medical conditions which caused these severe morbidities?

Figure 1 is very basic and could be strengthened by including bars comparing average PCMC scores for various subpopulations (the 3 sub categories can easily be shown in a stacked bar as part of the total)

How did 99% of women think they were treated poorly by health professions but 63% thought they were treated with respect - is this not measuring the same thing?

The percent of women delivering without a companion should probably be shown with the denominator of those who wanted to have a companion, as that would be a better proxy for respectful care.

What is the definition of "supportive care"?

How was "maternal outcome" assessed?

The discussion section could use more detail:

If many women had low PCMC scores but reported high satisfaction, does that indicate that expectations are low? Can this be unpacked - what is the literature on this about social forces that contribute to these expectations?

If women delivered by a nurse had better PCMC scores than those delivered by a midwife, what is the hypothesis about how nurses vs midwives are trained?

Why might practices be different vs day than night? Are there different staffing levels or cadres available?

Figures 3A, B, and C are too blurry to read.

Reviewers' comments:

Reviewer's Responses to Questions

**Comments to the Author**

1. Is the manuscript technically sound, and do the data support the conclusions?

Reviewer #1: Partly

Reviewer #2: Yes

2. Has the statistical analysis been performed appropriately and rigorously? 

Reviewer #1: Yes

Reviewer #2: Yes

3. Have the authors made all data underlying the findings in their manuscript fully available?

Reviewer #1: No

Reviewer #2: Yes

4. Is the manuscript presented in an intelligible fashion and written in standard English?

Reviewer #1: Yes

Reviewer #2: No

5. Review Comments to the Author

Reviewer #1: Overall, I think this is an area of needed research, especially understanding the intersecting relationships between person-centered maternity care, quality of clinical care, and women’s evaluations of the care experience. However, it appears that this study may need additional analyses, because I am not sure we are getting the full picture.

1. According to Chalmers & Porter (2003), the Bologna score quantifies “the extent to which labors have been managed as if normal as opposed to complicated.” Given that 51.8% of women had at least one risk factor and that 13.0% of women had at least one adverse outcome, please justify the use of this scale for this purpose among this sample.

2. Satisfaction scores are only informative to the degree that they indicate what women are satisfied with. What is the question used for “total satisfaction”? The frame of the question will help give context to what women were actually evaluating (e.g. was it satisfaction with care, the experience of childbirth, etc.?).

3. I think this study could benefit from further analysis.

a. First, descriptions and results of multivariate analysis could be better described and presented (e.g. what were the variables included in the final multivariate model? What are the estimates associated with PCMC or Bologna score in the multivariate models).

b. Though satisfaction was evaluated on a Likert scale, the multivariate analysis used logistic regression, splitting satisfaction into 6 and above versus under 6. It seems that linear regression seems more appropriate, assessing the incremental impact of the PCMC or Bologna scales.

c. Furthermore, the indicators included in the Bologna scale may not carry equal weight for women’s perceptions of care, especially because satisfaction may be based on women’s expectations and conditional on their social context. For instance, because induction of labor is commonly practiced and may be expected, it might not negatively impinge on women’s satisfaction. Perhaps assessments of individual indicators with women’s satisfaction might reveal the extent to which certain practices are correlated with satisfaction in this context.

d. One of the measures in the Bologna score was the presence of a labour companion, however many of the qualitative results (Supplement, Table 6) indicate that many women did not want a labor companion. In this case, the corresponding indicator within the Bologna scale would not represent better clinical care (for example, providers respecting a woman's decision to not include a companion would represent higher quality care). This should, in some way, be considered in your handling of the Bologna scale (and especially accounted for in your multivariate model).

Smaller issues:

For figure 1, scores are not easily comparable since they use different scales. It might be more helpful to display scores as percentages, so that they will be presented on the same scale.

Do you have information about what factors are associated with induction of labour, either qualitative or quantitative? Is it regular practice based on longer labours? Is it based on certain criteria of women’s conditions? Did they tend to be at night, etc.?

In the Supplementary Files, Table 5a-c, categorizes each measure into 3 groups. What is the reasoning behind these specific groupings? (For example, why is the first category for PCMC 0-58?)

Reviewer #2: This is an interesting article on an important and timely topic. Methods are clearly described, the authors used reliable study measures and are transparent about their protocols and procedures.

Overall the literature review is a bit thin – could use more of a rationale for why they chose to use a measure of satisfaction when for some time now researchers have understood satisfaction to be a poor discriminator of quality, in some LMICS, that their expectations for a low level of supportive care and the relief of having a live baby often leads to higher satisfaction scores even when they or observers report objective evidence of mistreatment. The discussion would also be enhanced with a bit more in depth exploration of some of the findings, as indicated in my notes below. In particular a discussion about the findings on quality of care as relates to global evidence on the overuse of interventions is missing – the authors limit their discussion to a small section on the components of the Bologna score without exploring the disconnect between overuse of interventions and patient satisfaction.

The manuscript is mostly well organized and written in acceptable English but there are issues with syntax, missing plurals, tense, and typographical errors throughout – Since PLOS does not copyedit before publishing, I strongly recommend the authors arrange for copyediting by a native English speaker who is a good editor before resubmission.

1. Some examples of English language errors:

• Notably, in Sri Lanka the maternal mortality rate had a major declined

over the last sixty years - it was 1694/100,000 in 1947 - to reach one of the lowest rate_ in the South Asian Region, despite Sri Lanka being a lower middle-income country [1]

• These remarkable achievements have been reached on the back of consistent commitment_ toward health and health-related policies, including as critical aspects (of?/as?) the provision of free of charge education and free of charge health services [3,4].

• For example, despite WHO explicitly recommends labor companionship as a low-cost intervention to improve outcomes of labor [16], and despite Sri Lankan government has explicitly included this in a national policy [17], a recent survey highlighted that nearly 60% of consultant obstetricians did not allow labour companions in their wards [18].

• Women who underwent a caesarian section, or with an age outside the inclusion criteria, or with major psychiatric illnesses, or hospitalized in intensive care unit, or refusing consent, were excluded.

• On the other side, the PCMC score significantly changed in different ethnic group, in women with more pregnancies, and by type of professionals that assisted the delivery.

2. There are now several studies exploring mistreatment and abuse of women during pregnancy globally – please specify if you are referring to South Asian studies….

“Although few studies have explored the area of mistreatment and abuse of women during pregnancy, existing qualitative reports suggest a tendency for discriminatory behavior (such as verbal, emotional and even sexual abuse) and a diffuse normalization of disrespectful and abusive treatment of female patients [19,20”

3. Please specify what type of ‘training” the researcher received:

“The questionnaire was administered in the immediate post-natal period, before discharge, by an independent trained researcher. “

4. Re the discussion about the use of partograph as an indicator of quality via the Bologna Score: The WHO no longer recommends the use of partograph as a measure of quality: See these articles by their team:

Bonet M, Oladapo OT, Souza JP, Gulmezoglu AM. Diagnostic accuracy of the partograph alert and action lines to predict adverse € birth outcomes: a systematic review. BJOG 2019;126:1524–1533.

Souza JP, Oladapo OT, Fawole B, Mugerwa K, Reis R, Barbosa-Junior F, Oliveira-Ciabati L, Alves D, G€ulmezoglu AM. Cervicaldilatation over time is a poor predictor of severe adverse birth outcomes: a diagnostic accuracy study. BJOG 2018;125:991–1000.

Please discuss the more current recommendations for monitoring, interpretation, and management of labour progress in light of your findings.

5. Please justify the rationale for recoding the Likert scale for satisfaction into a binary especially in light of the subtleties in using satisfaction as a measure of quality of experience:

“Women satisfaction was analyzed as a binary outcome (Likert scale equal or more than 6 versus Likert scale less than 6) and the odds ratio (OR) of each predictor on it was calculated through bivariate logistic regression.”

6. Please specify how the women were “ involved in the study by providing their views on the quality of care received.” Did they participate in survey development? Pilot test? Content Validate the measures?

7. Points that need more in depth Discussion:

• Despite the following interesting finding: “Nearly two thirds (61.7%) were assisted by a nurse, one third (33.7%) by a midwife, and only a minority by a doctor.”

there is almost no discussion about the differential effects of the type of provider on the quality of care (aside from noting women reported more respect by nurses than midwives) , nor explanation of potential reasons for these differences. This is important to unpack especially in light of global evidence that suggests that midwives provide more respectful care. Please also add some information in the background about the organization of care in Sri Lanka, the respective roles of providers, and describe the caseload vs service based models available.

• The mean PCMC score was significantly higher in Sinhalese women compared to Muslim (mean difference: 3.3; p=0.041) and to Tamil (mean difference: 3.8; p=0.049). S

This sentence and finding also deserves more attention in the discussion – please acknowledge this ethnic disparities in PCMC and address any implied or known cultural racism and bias that exists within the socio political climate, and contributes to these findings. This is not unlike other jurisdictions where marginalized populations experience more mistreatment (See Vedam et al. 2019, Giving Voice to Mothers, Reproductive Health).

8. Given the high rates of different types of mistreatment and violations of human rights reported, the emphasis in the following sentence appears misplaced. I suggest that the clause should begin with less than two thirds rather than nevertheless, and there should be some discussion about why this type of behavior was acceptable to those in the two thirds portion of the data.

“Notably, the majority of women (99.3%) reported to have been treated with an unfriendly

manner by health professionals, nevertheless about two thirds (63.5%) thought that medical

staff treated them with respect.”

9. Please take this opportunity discuss the following findings in light of global health human rights standards rather than deflecting this to a mandate for future study or simply development of courses to “promote PCMC”:

“Overall one out of six (14.8%) felt to have been treated roughly like pushed, beaten, slapped, pinched, physically restrained, or gagged. About one third (28.5%) reported to have been shouted, scolded, insulted, threatened, or talked to rudely. For

most women (85.8%) the health professionals did not explain the drugs given, and more than

half (55%) didn’t feel involved in decisions about their care, nor were asked for permission or

consent before performing procedures (57%). Less than a quarter (21.0%) thought that health

professionals took the best of care of them or did everything they could to help control their

pain (21.8%).

10. Please explain the following statements further – not clear as is:

• “Interestingly, women’s satisfaction had a very poor correlation with the Bologna score, but a moderate correlation with PCMC, suggesting that women’s satisfaction may have been more affected by the “experience of care” than by the “provision of care”, and that the two domains were very poorly interconnecting, in women’s views.”

• Notably, in this study in Sri Lanka some of items of the Bologna score actually indicated good practices, for example delivery in non-supine position was much more frequent tin this study than what reported in a study in Italy [35].

• On the other side, the PCMC score significantly changed in different ethnic group, in women with more pregnancies, and by type of professionals that assisted the delivery.

6. PLOS authors have the option to publish the peer review history of their article (what does this mean?). If published, this will include your full peer review and any attached files.

Reviewer #1: No

Reviewer #2: **Yes: **Saraswathi Vedam

---

## [Author Response · Author response to Decision Letter 0]

12 Nov 2020

CC: esacks@jhsph.edu

PONE-D-20-18688

Correlation among experience of person-centered maternity care, provision of care and women’s satisfaction: cross sectional study in Colombo, Sri Lanka

PLOS ONE

Dear Dr. Rishard,

Thank you for submitting your manuscript to PLOS ONE. After careful consideration, we feel that it has merit but does not fully meet PLOS ONE’s publication criteria as it currently stands. Therefore, we invite you to submit a revised version of the manuscript that addresses the points raised during the review process.

● A rebuttal letter that responds to each point raised by the academic editor and reviewer(s). You should upload this letter as a separate file labeled 'Response to Reviewers'.

● A marked-up copy of your manuscript that highlights changes made to the original version. You should upload this as a separate file labeled 'Revised Manuscript with Track Changes'.

● An unmarked version of your revised paper without tracked changes. You should upload this as a separate file labeled 'Manuscript'.

We look forward to receiving your revised manuscript.

Kind regards,

Emma Sacks

Academic Editor

PLOS ONE

Journal Requirements:

 content.

***The format of the manuscript has been revised following the above guidelines

 ***Thank you for the comment. We have provided additional details in the paper. We obtained ethical approval from the ethics review committee and permission from the director of De Soysa Hospital for women before conducting the interviews. We provided the informed consent form in Tamil, English and Singhala. Details regarding the research were provided by a research assistant who had prior experience in research and a degree in biomedical science. Informed written consent was taken from participants. In case of minors informed assent and consent from guardians / parents were obtained. 

***The Supporting Information files and their citation in the manuscript have been revised following the above guidelines

Additional Editor Comments (if provided):

BEFORE SUBMITTING A REVISION, PLEASE ADD PAGE AND LINE NUMBERS. IT IS VERY DIFFICULT FOR REVIEWERS TO GIVE COMMENTS WITHOUT THESE.

***Thank you for the suggestion. We revised the format of the manuscript 

While this is a critically important topic, and clearly understudied in Sri Lanka, this paper needs significantly more analytic work. The analyses are very basic and present only the totals, despite the methods indicating the ability to assess various correlations between demographics and practices, and between the PCMC, Bologna, and Likert scales. The results presented here are cursory. Detailed results should be in the paper, not supplementary appendices.

***Thank you for your comment, which gave us the opportunity to improve the analysis and to report relevant results in the main text. We have now moved some of the key results from the supplementary appendices to the manuscript, such as score distributions. Moreover, bivariate and multivariate analyses were performed and reported in the manuscript and a sensitivity analysis was conducted.

This paper requires more editing for grammar (even in the first paragraph, there are mistakes) and spelling (including Caesarean etc).

Please spell out acronyms, including PCMC, IUGR etc, at first use and in tables.

***Thank you for the comment. The grammar and spelling were revised and edited by a native English speaker. 

Please include in limitations that not including women with psychiatric illnesses may introduce bias, as these women may be more likely to experience mistreatment.

***Thank you for the comment. The limitation suggested has been included in line 3 of the last paragraph of the discussion. 

Please include information about how women below age 18 were consented/assented.

***Thank you for this observation, women below age 18 were consented /assented.

Please include more specifics in the methods - how were women recruited? what is an "established individual patient database" = are these hospital records?

***Thank you for your comment. We have clarified the procedure related to the collection on women characteristics and outcomes in the lines 148-150. This data collection process has been in practice since July 2015.

In relation to the other indicators, eligible mothers were identified using the birth registry: all the consecutive deliveries were approached. All eligible women who consented to participate in the study were interviewed. Deliveries that took place during daytime, nights and weekends were included. 

Please include more details about the sample size calculation.

***Thank you for the comment which gave us the opportunity to better clarify the methods and to include specific details as suggested.

Please explain how it was ensured that women understood that research assistants were not hospital staff (which may have led to desirability bias) 

 ***Desirability bias was minimised by having an interviewer who is not a staff member. Both the interviewers’ outfit/uniform and the identification card clearly allowed them to be identified as non-staff members. Beside this, interviewers introduced themself and their role. Furthermore, they clarified that interviews were anonymous. Interviews were conducted in a separate area with privacy. 

Why were indicators chosen like "non supine" rather than "women's preferred birthing positions were respected"? Wouldn't that be more person-centered?

***Thank you for your observation. We very much agree with you in principle, but we had to adhere to a predefined scoring system. In fact, we followed the definitions provided by the Bologna Score System, which uses delivery in non-supine as the indicator to measure this practice. Women’s preferences in birthing positions will be encouraged during the future interventions to improve quality of care. 

When referring to "risk factors" please clarify "medical risk factors" (to differentiate from social risk)

***Thank you for the suggestion. The text was revised accordingly.

"Near miss" is vague for a medical event - can you include the medical conditions which caused these severe morbidities?

***Thank you for the comment. The definition of “maternal near miss” was added in the notes of table 1. The definition was agreed locally. It is based on the recommendation of the WHO Manual “Evaluating the quality of care for severe pregnancy complications: the WHO near-miss approach for maternal health.” https://apps.who.int/iris/bitstream/handle/10665/44692/9789241502221_eng.pdf;jsessionid=4BB1351F7F82F8B755AA9220A11279B4?sequence=1

Figure 1 is very basic and could be strengthened by including bars comparing average PCMC scores for various subpopulations (the 3 sub categories can easily be shown in a stacked bar as part of the total)

***Thank you for the comment. We improved the figure, which at the moment is figure 2, showing each component of the PCMC score normalised to 100 to allow a better comparison among full score and subscores. We also inserted another figure (which in the current manuscript is Fig 1) where we reported the distribution of PCMC, Bologna score and women’s satisfaction Likert scale

How did 99% of women think they were treated poorly by health professions but 63% thought they were treated with respect - is this not measuring the same thing?

*** Thank you for your observation. The PCMC questionnaire includes the following two questions: 1) Did the doctors, nurses, or other staff at the facility treat you with respect? 2) Did the doctors, nurses, and other staff at the facility treat you in a friendly manner?

These questions focus on related yet different aspects of respectful care. Results support the hypothesis that not being treated in a friendly manner is not perceived to be a discriminating component for respectful care. 

The percent of women delivering without a companion should probably be shown with the denominator of those who wanted to have a companion, as that would be a better proxy for respectful care.

*** Thank you for your suggestion. We decided to show percentages of reasons for absence of a companion at the time of birth with the number of women without a companion as denominator to give emphasis to the impact of socio-cultural context on women’s preferences (please see Table 2).

What is the definition of "supportive care"? How was "maternal outcome" assessed?

***Thank you for the question. The PCMC questionnaire includes three subdomains, the specification of these has been inserted in table S4. One of these is called “Supportive care”. This subdomain includes 15 items which measure supportive care and are related to the waiting time, the pain control, the companion during delivery, the hospital environment. We provide hereby a reference “Afulani PA, Phillips B, Aborigo RA, Moyer CA. Person-centred maternity care in low-income and middle-income countries: analysis of data from Kenya, Ghana, and India. Lancet Glob Health. 2019 Jan;7(1):e96-e109. doi: 10.1016/S2214-109X(18)30403-0.”

For maternal outcomes we intended maternal health outcomes. we have clarified this better in the text

The discussion section could use more detail:

If many women had low PCMC scores but reported high satisfaction, does that indicate that expectations are low? Can this be unpacked - what is the literature on this about social forces that contribute to these expectations?

***Thank you for your comment, which gave us the opportunity to add an explanation and additional references in the discussion section. 

Although the PCMC tool focuses on the experience of care and includes different dimensions such as communication, respect, dignity, and emotional support, which are highlighted as key dimensions also in the World Health Organization (WHO) quality of care framework for maternal and newborn health (Tunçalp et al, BJOG 2015), women satisfaction may be depended on many other factors. We report below what literature reports as key aspects that contribute to these expectations

● Women construct their birth experience differently. Views are directed by personal beliefs and values (Martin CH, Fleming V. The birth satisfaction scale. Int J Health Care Qual Assur. 2011;24(2):124-35. doi: 10.1108/09526861111105086. PMID: 21456488)

● Ethnicity, religion, facility location have all been shown to influence women’s satisfaction. (Srivastava, A., Avan, B.I., Rajbangshi, P. et al. Determinants of women’s satisfaction with maternal health care: a review of literature from developing countries. BMC Pregnancy Childbirth 15, 97 (2015). https://doi.org/10.1186/s12884-015-0525-0

● It has been described that women tend to normalize the disrespectful care when they experience good outcomes (Freedman LP, Kruk ME. Disrespect and abuse of women in childbirth: challenging the global quality and accountability agendas. Lancet. 2014;384:e42–4 and Bowser D, Hill K. Exploring evidence and abuse in facility-based childbirth: report of a landscape analysis. Washington DC: USAID-TRAction Project; 2010. Available from: http://wwwtractionprojectorg/sites/default/files/ Respectful_Care_at_Birth_9-20-101_Final.pdf) 

Our findings of high satisfaction rate could be due to perception of satisfaction that women generally feel after the arrival of their newborns, in a context where this is the priority value.

If women delivered by a nurse had better PCMC scores than those delivered by a midwife, what is the hypothesis about how nurses vs midwives are trained?

***Thank you for the comment. We have further checked this variable. In real practice nurses and midwives work as members. Mothers usually receive care from both cadres at different times. In addition, mothers may find difficult to identify correctly nurses from midwife. For these reasons, we decided to exclude the variable related to the person who delivered from the analysis. 

Why might practices be different vs day than night? Are there different staffing levels or cadres available?

***Thank you for the comment. In many settings staffing levels are lower at night. In the study setting staff allocation at night is less. Onsite consultant is also available only during daytime, while during nights is “on call”. 

Figures 3A, B, and C are too blurry to read.

*** Thank you for the comment. The figures were revised.

Reviewers' comments:

Reviewer's Responses to Questions

Comments to the Author

1. Is the manuscript technically sound, and do the data support the conclusions?

Reviewer #1: Partly

Reviewer #2: Yes

2. Has the statistical analysis been performed appropriately and rigorously?

Reviewer #1: Yes

Reviewer #2: Yes

3. Have the authors made all data underlying the findings in their manuscript fully available?

Reviewer #1: No

Reviewer #2: Yes

4. Is the manuscript presented in an intelligible fashion and written in standard English?

Reviewer #1: Yes

Reviewer #2: No

5. Review Comments to the Author

Reviewer #1: Overall, I think this is an area of needed research, especially understanding the intersecting relationships between person-centered maternity care, quality of clinical care, and women’s evaluations of the care experience. However, it appears that this study may need additional analyses, because I am not sure we are getting the full picture.

***Thank you for your appreciation.

1. According to Chalmers & Porter (2003), the Bologna score quantifies “the extent to which labors have been managed as if normal as opposed to complicated.” Given that 51.8% of women had at least one risk factor and that 13.0% of women had at least one adverse outcome, please justify the use of this scale for this purpose among this sample.

***Thank you for your comment. Chalmers & Porter (2003) stated that this tool attempts to quantify to which extend the labour is managed normally as opposed to complicated labour. In terms of labour, all women included in our study had a physiological labour. 

To our knowledge having a medical risk factor (such as diabetes or hypertension) or having, after physiological labour, an adverse health outcome is not a criterion of exclusion for using the Bologna Score. 

This tool was designed to use to study the behaviours and practices towards normal labour care within the maternity services. Further this tool can be used to assess and compare the childbirth support process and practice of evidence-based care throughout the world.

(BIRTH 35:4 December 2008 321 Care in Labor: A Swedish Survey Using the Bologna Score Ann-Kristin Sandin-Bojo ¨, RN, RM, PhD, and Linda J. Kvist, RN, RM, PhD)

2. Satisfaction scores are only informative to the degree that they indicate what women are satisfied with. What is the question used for “total satisfaction”? The frame of the question will help give context to what women were actually evaluating (e.g. was it satisfaction with care, the experience of childbirth, etc.?).

*** The question was phrased as follows “What is your overall satisfaction with care received, on a scale 1 (minimum )to 10 (maximum)? 

Several other studies used this direct simple question to assess the overall degree of satisfaction of mothers. Many studies also use this phrasing to assess service users satisfaction with care received.

References: 

1. Lazzerini M, Mariani I, Semenzato C, Valente EP. Association between maternal satisfaction and other indicators of quality of care at childbirth: a cross-sectional study based on the WHO standards. BMJ Open. 2020 Sep 14;10(9):e037063. doi: 10.1136/bmjopen-2020-037063. PMID: 32928854; PMCID: PMC7490935.

2. Conesa Ferrer MB, Canteras Jordana M, Ballesteros Meseguer C, et al. Comparative study analysing women's childbirth satisfaction and obstetric outcomes across two different models of maternity care. BMJ Open 2016;6:e011362. 10.1136/bmjopen-2016-011362.

3. Alfaro Blazquez R, Corchon S, Ferrer Ferrandiz E. Validity of instruments for measuring the satisfaction of a woman and her partner with care received during labour and childbirth: systematic review. Midwifery 2017;55:103–12. 10.1016/j.midw.2017.09.014.

3. I think this study could benefit from further analysis.

a. First, descriptions and results of multivariate analysis could be better described and presented (e.g. what were the variables included in the final multivariate model? What are the estimates associated with PCMC or Bologna score in the multivariate models).

***Thank you for your suggestion, which give us the opportunity to improve the paper we added a table “Table 2. Two- and multi-ways ANOVA” in the manuscript with the results of the analysis as suggested.

b. Though satisfaction was evaluated on a Likert scale, the multivariate analysis used logistic regression, splitting satisfaction into 6 and above versus under 6. It seems that linear regression seems more appropriate, assessing the incremental impact of the PCMC or Bologna scales.

***Thank you for your comment. We opted for a logistic regression model based on the fact that the assumptions for a linear regression models did not hold (eg, normality) therefore both linear regression estimates and relative statistic tests could be biased. The cut off at 6 was taken as the minimum satisfaction limit. Moreover, as described in the method section, a sensitivity analysis was performed dichotomizing satisfaction Likert scale at its median value. Results of the sensitivity analysis are shown in S8 Table and reported in the result section in the paragraph “Univariate and multivariate analyses” (lines 367-371).

c. Furthermore, the indicators included in the Bologna scale may not carry equal weight for women’s perceptions of care, especially because satisfaction may be based on women’s expectations and conditional on their social context. For instance, because induction of labor is commonly practiced and may be expected, it might not negatively impinge on women’s satisfaction. Perhaps assessments of individual indicators with women’s satisfaction might reveal the extent to which certain practices are correlated with satisfaction in this context.

***Thank you for your comment. We fully agree. We revised the paper and assessed the association between each component of the Bologna score with women’s satisfaction including them in the bivariate and multivariate analysis. 

d. One of the measures in the Bologna score was the presence of a labour companion, however many of the qualitative results (Supplement, Table 6) indicate that many women did not want a labor companion. In this case, the corresponding indicator within the Bologna scale would not represent better clinical care (for example, providers respecting a woman's decision to not include a companion would represent higher quality care). This should, in some way, be considered in your handling of the Bologna scale (and especially accounted for in your multivariate model).

*** Thank you for your suggestion which gave us the opportunity to better clarify this point in the paragraph “Provision of care” in the result section. We have refined our analysis, based also on the recommendation of another review, in order to report these data in more detail (Figure 3, table 2, and text lines 316-319). 

Smaller issues:

For figure 1, scores are not easily comparable since they use different scales. It might be more helpful to display scores as percentages, so that they will be presented on the same scale.

***Thank you for your suggestion. To allow an easy comparison across the different PCMC domains, rescaled scores were calculated as the fraction of the total possible score on each domain and normalised to 100 (in line with what done by Afulani, Lancet global health 2020).

Do you have information about what factors are associated with induction of labour, either qualitative or quantitative? Is it regular practice based on longer labours? Is it based on certain criteria of women’s conditions? Did they tend to be at night, etc.?

*** Thank you for your suggestion. We strongly believe that it would be beneficial for our setting to address the Induction of labour (IOL) in a separate paper, which indeed has already been planned. We are in the process of conducting a comprehensive audit of IOL practices over 4 years, to identify what factors would have led to high rates of IOL. This separate paper shall be submitted soon.

In the Supplementary Files, Table 5a-c, categorizes each measure into 3 groups. What is the reasoning behind these specific groupings? (For example, why is the first category for PCMC 0-58?)

***Thank you for the comment. Based on referee’s comments we decided to represent the table previously listed as Table 5a-c of Supplementary Files differently, and now it became figure 1. In this Figure 1 we reported the distributions of the scores through histograms and estimated normal distributions for subpopulations which resulted statistically significant in the bivariate analysis.

Reviewer #2: This is an interesting article on an important and timely topic. Methods are clearly described, the authors used reliable study measures and are transparent about their protocols and procedures.

***Thank you for your appreciation.

Overall the literature review is a bit thin – could use more of a rationale for why they chose to use a measure of satisfaction when for some time now researchers have understood satisfaction to be a poor discriminator of quality, in some LMICS, that their expectations for a low level of supportive care and the relief of having a live baby often leads to higher satisfaction scores even when they or observers report objective evidence of mistreatment. The discussion would also be enhanced with a bit more in depth exploration of some of the findings, as indicated in my notes below. In particular a discussion about the findings on quality of care as relates to global evidence on the overuse of interventions is missing – the authors limit their discussion to a small section on the components of the Bologna score without exploring the disconnect between overuse of interventions and patient satisfaction.

***Thank you for your comment and suggestion which gave us the opportunity to refine the literature and the discussion.

We report hereby the references added:

● Ghana

Dzomeku, V.M., Boamah Mensah, A.B., Nakua, E.K., Agbadi P, Lori JR, Donkor P. “I wouldn’t have hit you, but you would have killed your baby:” exploring midwives’ perspectives on disrespect and abusive Care in Ghana. BMC Pregnancy Childbirth 20, 15 (2020).

● Iran

Hajizadeh, K., Vaezi, M., Meedya, S., Charandabi ASM, Mirghafourvand M. Prevalence and predictors of perceived disrespectful maternity care in postpartum Iranian women: a cross-sectional study. BMC Pregnancy Childbirth 20, 463 (2020). https://doi.org/10.1186/s12884-020-03124-2

● South African

Lappeman, M., Swartz, L. Rethinking obstetric violence and the “neglect of neglect”: the silence of a labour ward milieu in a South African district hospital. BMC Int Health Hum Rights 19, 30 (2019). https://doi.org/10.1186/s12914-019-0218-2

Sharma, G., Penn-Kekana, L., Halder, K. Filippi V. An investigation into mistreatment of women during labour and childbirth in maternity care facilities in Uttar Pradesh, India: a mixed methods study. Reprod Health 16, 7 (2019). https://doi.org/10.1186/s12978-019-0668-y

● Ethiopia

Kassa, Z.Y., Husen, S. Disrespectful and abusive behavior during childbirth and maternity care in Ethiopia: a systematic review and meta-analysis. BMC Res Notes 12, 83 (2019). https://doi.org/10.1186/s13104-019-4118-2

Gebremichael, M.W., Worku, A., Medhanyie, Edin AAK, Berhane Y. Women suffer more from disrespectful and abusive care than from the labour pain itself: a qualitative study from Women’s perspective. BMC Pregnancy Childbirth 18, 392 (2018). https://doi.org/10.1186/s12884-018-2026-4

Ukke GG, Gurara MK, Boynito WG. Disrespect and abuse of women during childbirth in public health facilities in Arba Minch town, south Ethiopia - a cross-sectional study. PLoS One. 2019 Apr 29;14(4):e0205545. doi: 10.1371/journal.pone.0205545. PMID: 31034534; PMCID: PMC6488058.

● Tanzania

Kruk ME, Kujawski S, Mbaruku G, Ramsey K, Moyo W, Freedman LP. Disrespectful and abusive treatment during facility delivery in Tanzania: a facility and community survey. Health Policy Plan. 2018 Jan 1;33(1):e26-e33. doi: 10.1093/heapol/czu079. PMID: 29304252.

● Brazil

Lansky S, Souza KV, Peixoto ERM, Oliveira BJ, Diniz CSG, Vieira NF, et al. Obstetric violence: influences of the Senses of Birth exhibition in pregnant women childbirth experience. Cien Saude Colet. 2019 Aug 5;24(8):2811-2824. Portuguese, English. doi: 10.1590/1413-81232018248.30102017. PMID: 31389530.

● Nigeria

Ishola F, Owolabi O, Filippi V. Disrespect and abuse of women during childbirth in Nigeria: A systematic review. PLoS One. 2017 Mar 21;12(3):e0174084. doi: 10.1371/journal.pone.0174084. PMID: 28323860; PMCID: PMC5360318. 

● Myanmar

Maung, T.M., Show, K.L., Mon, N.O. et al. A qualitative study on acceptability of the mistreatment of women during childbirth in Myanmar. Reprod Health 17, 56 (2020). https://doi.org/10.1186/s12978-020-0907-2

Additional references:

Bohren MA, Mehrtash H, Fawole B, Maung TM, Balde MD, Maya E, et al. How women are treated during facility-based childbirth in four countries: a cross-sectional study with labour observations and community-based surveys. Lancet. 2019 Nov 9;394(10210):1750-1763. doi: 10.1016/S0140-6736(19)31992-0. Epub 2019 Oct 8. PMID: 31604660; PMCID: PMC6853169.

Brizuela V, Leslie HH, Sharma J, Langer A, Tunçalp Ö. Measuring quality of care for all women and newborns: how do we know if we are doing it right? A review of facility assessment tools. Lancet Glob Health. 2019;7:e624–e632

The manuscript is mostly well organized and written in acceptable English but there are issues with syntax, missing plurals, tense, and typographical errors throughout – Since PLOS does not copyedit before publishing, I strongly recommend the authors arrange for copyediting by a native English speaker who is a good editor before resubmission.

***Thank you for your suggestion. The manuscript was revised and edited by a native speaker

1. Some examples of English language errors:

• Notably, in Sri Lanka the maternal mortality rate had a major declined

over the last sixty years - it was 1694/100,000 in 1947 - to reach one of the lowest rate_ in the South Asian Region, despite Sri Lanka being a lower middle-income country [1]

• These remarkable achievements have been reached on the back of consistent commitment_ toward health and health-related policies, including as critical aspects (of?/as?) the provision of free of charge education and free of charge health services [3,4].

***The manuscript was revised and edited.

• For example, despite WHO explicitly recommends labor companionship as a low-cost intervention to improve outcomes of labor [16], and despite Sri Lankan government has explicitly included this in a national policy [17], a recent survey highlighted that nearly 60% of consultant obstetricians did not allow labour companions in their wards [18].

***The manuscript was revised and edited.

• Women who underwent a caesarian section, or with an age outside the inclusion criteria, or with major psychiatric illnesses, or hospitalized in intensive care unit, or refusing consent, were excluded.

*** The manuscript was revised and edited.

• On the other side, the PCMC score significantly changed in different ethnic group, in women with more pregnancies, and by type of professionals that assisted the delivery.

*** The manuscript was revised and edited.

2. There are now several studies exploring mistreatment and abuse of women during pregnancy globally – please specify if you are referring to South Asian studies….

“Although few studies have explored the area of mistreatment and abuse of women during pregnancy, existing qualitative reports suggest a tendency for discriminatory behavior (such as verbal, emotional and even sexual abuse) and a diffuse normalization of disrespectful and abusive treatment of female patients [19,20”

***Thank you for your comment. Indeed we were referring to south Asia, we specified it accordingly in the manuscript. 

3. Please specify what type of ‘training” the researcher received:

“The questionnaire was administered in the immediate post-natal period, before discharge, by an independent trained researcher. “

***Thank you for the comment. The trained researcher had prior experience in patient interviewing. She was trained directly by the principal investigator on all standard operating procedures (SOP), which included; women’s eligibility criteria, how to approach mothers, privacy, data quality checks etc. She was supervised high-intensity in the field for a week in which she improved the skills in taking informed consent, interviewing, and recording data. Once the primary investigator felt that all queries were solved, she started data collection. 

4. Re the discussion about the use of partograph as an indicator of quality via the Bologna Score: The WHO no longer recommends the use of partograph as a measure of quality: See these articles by their team:

Bonet M, Oladapo OT, Souza JP, Gulmezoglu AM. Diagnostic accuracy of the partograph alert and action lines to predict adverse birth outcomes: a systematic review. BJOG 2019;126:1524–1533.

Souza JP, Oladapo OT, Fawole B, Mugerwa K, Reis R, Barbosa-Junior F, Oliveira-Ciabati L, Alves D, G€ulmezoglu AM. Cervicaldilatation over time is a poor predictor of severe adverse birth outcomes: a diagnostic accuracy study. BJOG 2018;125:991–1000.

Please discuss the more current recommendations for monitoring, interpretation, and management of labour progress in light of your findings.

*** We have revised literature and we got in touch with WHO staff at head quarter. 

● The paper that you refer to (Bonet M, Oladapo OT, Souza JP, Gulmezoglu AM. Diagnostic accuracy of the partograph alert and action lines to predict adverse birth outcomes: a systematic review. BJOG 2019;126:1524–1533.) assessed one specific parameter of the partograph, and concluded that “This systematic review does not support the use of the cervical dilatation over time (at a threshold of 1 cm/h during active first stage) to identify women at risk of adverse birth outcomes.” However, authors suggest that care providers should continue to use other partograph parameters to monitor the well‐being of the woman and her baby, and identify risks for adverse birth outcomes and highlight the need of new tools to improve birth outcomes and reduce of unnecessary interventions during labour. The WHO recommendation is in line with this https://extranet.who.int/rhl/topics/preconception-pregnancy-childbirth-and-postpartum-care/care-during-childbirth/care-during-labour-1st-stage/who-recommendation-progress-first-stage-labour-diagnostic-test-accuracy-1%E2%80%93cm/hour-cervical

● We could not retrieve any formal statement from WHO where the use of partogram is NOT recommended. 

● Sri Lankan guidelines for maternal care has emphasized the use of partogram http://fhb.health.gov.lk/index.php/en/technical-units/maternal-care-unit

5. Please justify the rationale for recoding the Likert scale for satisfaction into a binary especially in light of the subtleties in using satisfaction as a measure of quality of experience:

“Women satisfaction was analyzed as a binary outcome (Likert scale equal or more than 6 versus Likert scale less than 6) and the odds ratio (OR) of each predictor on it was calculated through bivariate logistic regression.”

***Thank you for your comment. We opted for a logistic regression model due to a deviation from the assumptions of linear regression models (ie, normality) therefore both linear regression estimates and relative statistic tests could be biased. The cut off at 6 was taken as the minimum satisfaction limit. Moreover, as described in the method section, a sensitivity analysis was performed dichotomizing satisfaction Likert scale at its median value. Results of the sensitivity analysis are shown in S8 Table and reported in the result section in the paragraph “Univariate and multivariate analyses” (lines 367-371).

6. Please specify how the women were “ involved in the study by providing their views on the quality of care received.” Did they participate in survey development? Pilot test? Content Validate the measures?

***Thank you for the comment. Women participated in the tool development and validation. 

The tool has been validated in India and in other low middle-income countries where women had participated in the validation process. In our study, we included women in translation and adaptation process.

Four women Tamil speaking and four Sinhala speaking women were included in the expert committee, they examined the concordance with the original version. Two rounds of modifications were conducted and inadequate expressions/concepts of the translation, as well as any discrepancies, were corrected.

Further, 10 women who recently delivered, belonging to all age groups and from different socioeconomic backgrounds, were involved in pilot testing and content validity to check the understandability, clarity and acceptability of each tool separately. 

Series of focus group discussion have been done to obtain women’s views on quality of care and that will be discussed in a different paper. 

7. Points that need more in depth Discussion:

• Despite the following interesting finding: “Nearly two thirds (61.7%) were assisted by a nurse, one third (33.7%) by a midwife, and only a minority by a doctor.”there is almost no discussion about the differential effects of the type of provider on the quality of care (aside from noting women reported more respect by nurses than midwives) , nor explanation of potential reasons for these differences. This is important to unpack especially in light of global evidence that suggests that midwives provide more respectful care. Please also add some information in the background about the organization of care in Sri Lanka, the respective roles of providers, and describe the caseload vs service based models available.

***Based on comments from other referees we realized that this variable may not be accurate. Mothers are most often assisted by a team of people. They may have difficulties to distinguishing between a midwife and a nurse. For these reasons we have excluded this variable from the analysis. 

• The mean PCMC score was significantly higher in Sinhalese women compared to Muslim (mean difference: 3.3; p=0.041) and to Tamil (mean difference: 3.8; p=0.049). S

This sentence and finding also deserves more attention in the discussion – please acknowledge this ethnic disparities in PCMC and address any implied or known cultural racism and bias that exists within the socio political climate, and contributes to these findings. This is not unlike other jurisdictions where marginalized populations experience more mistreatment (See Vedam et al. 2019, Giving Voice to Mothers, Reproductive Health).

***Thank you for your comments. We agree with your observation. We added this additional point and references in the discussion section. 

8. Given the high rates of different types of mistreatment and violations of human rights reported, the emphasis in the following sentence appears misplaced. I suggest that the clause should begin with less than two thirds rather than nevertheless, and there should be some discussion about why this type of behavior was acceptable to those in the two thirds portion of the data.

“Notably, the majority of women (99.3%) reported to have been treated with an unfriendly manner by health professionals, nevertheless about two thirds (63.5%) thought that medical staff treated them with respect.”

***Thanks for this comment. In phrasing these results we did not want to sound judgmental. We have revised the sentence accordingly. 

Treating patients in a friendly manner is a psychosocial skill that health professionals should possess, however a metanalysis pointed out the many underlying causes at different levels (individual, hospital, national level) which can affect these behaviours (Mannava P, Durrant K, Fisher J, Chersich M, Luchters S. Attitudes and behaviours of maternal health care providers in interactions with clients: a systematic review. Global Health. 2015 Aug 15;11:36).

9. Please take this opportunity discuss the following findings in light of global health human rights standards rather than deflecting this to a mandate for future study or simply development of courses to “promote PCMC”:

“Overall one out of six (14.8%) felt to have been treated roughly like pushed, beaten, slapped, pinched, physically restrained, or gagged. About one third (28.5%) reported to have been shouted, scolded, insulted, threatened, or talked to rudely. For

most women (85.8%) the health professionals did not explain the drugs given, and more than half (55%) didn’t feel involved in decisions about their care, nor were asked for permission or consent before performing procedures (57%). Less than a quarter (21.0%) thought that health professionals took the best of care of them or did everything they could to help control their pain (21.8%).

***Thank you for this comment. We agree with your observation. We have now given more emphasis to this point, citing reference documents related to human rights, as well as the WHO statement on disrespect and abuse during childbirth at facility level. 

10. Please explain the following statements further – not clear as is:

• “Interestingly, women’s satisfaction had a very poor correlation with the Bologna score, but a moderate correlation with PCMC, suggesting that women’s satisfaction may have been more affected by the “experience of care” than by the “provision of care”, and that the two domains were very poorly interconnecting, in women’s views.”

***Thank you for the suggestion. We have explained this statement further. Bologna score focuses on “practices” while the PCMC focuses on the “experience “of care. In many settings these two aspect are not at equal quality. This emphasizes that improving good practices (as requested by the evidenced-based movement) without attention to the “relation” is not enough. 

• Notably, in this study in Sri Lanka some of items of the Bologna score actually indicated good practices, for example delivery in non-supine position was much more frequent tin this study than what reported in a study in Italy [35].

***We have explained this statement further. Large use of CTG during delivery in Italy may explain this difference. 

• On the other side, the PCMC score significantly changed in different ethnic group, in women with more pregnancies, and by type of professionals that assisted the delivery.

***We have explained this statement further. 

6. PLOS authors have the option to publish the peer review history of their article (what does this mean?). If published, this will include your full peer review and any attached files.

Do you want your identity to be public for this peer review? For information about this choice, including consent withdrawal, please see our Privacy Policy.

Reviewer #1: No

Reviewer #2: Yes: Saraswathi Vedam

---

## [Decision Letter · Decision Letter 1]

28 Jan 2021

PONE-D-20-18688R1

Correlation among experience of person-centered maternity care, provision of care and women’s satisfaction: cross sectional study in Colombo, Sri Lanka

PLOS ONE

Dear Dr. Rishard,

Thank you for submitting your manuscript to PLOS ONE. After careful consideration, we feel that it has merit but does not fully meet PLOS ONE’s publication criteria as it currently stands. Therefore, we invite you to submit a revised version of the manuscript that addresses the points raised during the review process.

The reviewers acknowledge that the paper has been greatly improved by the revisions. They raise some additional important points. Please address these in a second revision. 

We look forward to receiving your revised manuscript.

Kind regards,

Tanya Doherty, PhD

Academic Editor

PLOS ONE

Reviewers' comments:

Reviewer's Responses to Questions

**Comments to the Author**

1. If the authors have adequately addressed your comments raised in a previous round of review and you feel that this manuscript is now acceptable for publication, you may indicate that here to bypass the “Comments to the Author” section, enter your conflict of interest statement in the “Confidential to Editor” section, and submit your "Accept" recommendation.

Reviewer #1: All comments have been addressed

Reviewer #2: (No Response)

2. Is the manuscript technically sound, and do the data support the conclusions?

Reviewer #1: Yes

Reviewer #2: Yes

3. Has the statistical analysis been performed appropriately and rigorously? 

Reviewer #1: Yes

Reviewer #2: Yes

4. Have the authors made all data underlying the findings in their manuscript fully available?

Reviewer #1: Yes

Reviewer #2: Yes

5. Is the manuscript presented in an intelligible fashion and written in standard English?

Reviewer #1: Yes

Reviewer #2: Yes

6. Review Comments to the Author

Reviewer #1: Thank you for your revision. The revised analyses and substantial discussion have greatly improved the paper. I especially appreciated the more detailed discussion surrounding women’s perceptions and normalization of disrespectful treatment.

You addressed some limitations with satisfaction as a measure, but there are other biases and limitations that should be addressed regarding satisfaction. For example, using satisfaction as a dichotomous variable, splitting satisfaction at 6 and above, will pose problems especially because respondents will often select the mid-point option as a cognitive bias. It appears that this is bias was also present in your sample, since over 15% of your sample selected 5. This suggests that a large fraction of those who selected the middle option were subject to cognitive biases and not influenced by other risk factors and likely overestimates the proportion of those who were “not satisfied.”

Rodway, P., Schepman, A., and Lambert, J. (2012). Preferring the One in the Middle: Further Evidence for the Centre-stage Effect. Applied Cognitive Psychology, 26 (2), 215-222 DOI: 10.1002/acp.1812

Your discussion could also benefit by addressing limitations of the Bologna score. I am not sure that it can be considered a measure of “the provision of care” as defined by the WHO QOC framework for maternal and newborn health. Perhaps it might be better described as “recommended clinical practices” or something that highlights clinical practices. Both companion of choice and delivery in the position of choice are considered to be respectful care practices and I believe are classified under the “experience of care.” Because of this, I do not think the Bologna score can be considered an appropriate measure of the “provision of care.”

Additionally, it should be mentioned that the correlation between PCMC and Bologna care should be interpreted conservatively, since it may be partially due to overlapping items. For example, the PCMC scale has items relating to companionship during labor and delivery, and whether the woman was able to deliver in the position she wanted.

Minor points:

Please address misspelled terms in footnotes of Table 1: “caesarian” “nera” “lapartotomy”

Line 304: “One third (28.5%)…” It seems that ‘over one-quarter’ or ‘nearly one-third’ would be more accurate than ‘one-third.’

Line 454: “heart” is misspelled

The figures are quite blurry. I am not sure if this because the images are of low resolution or if the system has distorted images.

Figure 3, please correct “Skin-Skin”

Figure 4, especially because some of your graphs include plots using only discrete values (i.e. 4b), please consider illustrating density in your plots (for example, using “jitter” or dodging points).

Reviewer #2: Thank you for the opportunity to read this revised version. The paper is greatly improved in clarity and impact as a result of the edits and additions made in response to reviewers’ comments. My remaining suggestions are mostly very minor copyediting recommendations; however, I do have one important note to the authors and editor:

This study adds to the growing body of literature that shows mistreatment and human rights violations during childbearing is a global and widespread phenomena (1/6), and is even more prevalent among historically disenfranchised communities (2/3). It is an important contribution to our understanding not only frequency of disrespect and abuse, but that these person-centred metrics of quality care are important outcome in their own right (eg. They do not need to be linked to other maternal newborn adverse outcomes, or patient satisfaction to merit urgent attention). Yet, the authors appear to be hesitant to let the data speak to this mandate, reserving the emphasis in the discussion solely to the key metric they chose to use: patient satisfaction. Perhaps they could at least state more clearly in their conclusions (abstract and paper) that “findings indicate evidence of poor quality care across several domains of mistreatment in childbirth in Sri Lanka. [and] Patient satisfaction as an indicator of quality care is inadequate to inform health systems reform”.

I can understand a reluctance to speak of prevalence of the most adverse outcomes (mistreatment) based on a sample size of 400 in the region (authors frame this analysis as pilot data), and I agree that discussions generally focus on findings per the originally intended methods. However, when results show egregious harm, in any study, it seems that researchers are obligated to name and elevate these findings and the urgency of rectifying harm and future targeted research on these matters. This is especially true when the findings align with incidence reported other published literature on the subject as cited.

Copy edits and typographical errors

Line 66 – Keywords - Suggested these edits to maximize searches for this article: Quality care, respectful maternity care, person-centered/person-centred, mistreatment, childbirth, disrespect and abuse (researchers in this field are unlikely to see satisfaction as an important component to include as noted by reviewer comments)

Line 96 (comma instead of full stop)

109-110 missing word: also defined [as] patient-centred, …

Line 142 suggest using “births” instead of deliveries – fast becoming the norm for respectful language (eg women give birth, providers

Line 293-296 The following sentence is unclear – either there were significant differences with some factors having lower scores or the scores were significantly lower for xxxx than xxxx

Additionally, other differences among the

294 full PCMC score (47.1; 95%CI 45.9-48.2) and “dignity and respect” (57.2; 95%CI 55.8-58.6), and

295 “supportive-care” subscores (50.5; 95%CI 49.0-51.9) were significantly lower (adjusted

296 p≤0.002 for all comparisons). PCMC not rescaled values are reported in Tables S2 and S3.

Line 303 awkward phrasing: 303 six (14.8%) felt to have been treated roughly – perhaps change to one out of six (14.8%) reported that they were treated roughly --or—one out of six (14.8%) reported mistreatment – all of the listed factors constitute mistreatment by global definitions.

Line 305 – best not to start a sentence with a percentage – perhaps rephrase: Most women (85.8%) reported that….

Line 357 women “at” their second pregnancy should be in their second pregnancy women….. or women [ in] their second pregnancy

422 mis-treated should be mistreated

Other comments

Line 368 “No factor was associated with women’s satisfaction” (sensitivity analysis just parity and non-supine position) -- this result shows that satisfaction is a poor discriminator of quality care and that finding should be discussed – with recommendation that future studies report on PCMC scores, and incidence of mistreatment as a stand alone indicator of quality, safety, rights.

Line in 394-399 that discusses above is too mild given the findings that 1/6 to ½ experience serious mistreatment – as written the discussion does not address that only one modifiable experience of care factor (birth position) was even moderately correlated – hence satisfaction is not measuring quality or the correlation shou

Lines 472-476 should immediately follow Line 427 as finding on differential experience by ethnicity are also socio-demographic predisposing factors. Then “ethnicity” should be added to the line 431 …-and better document how education, [ethnicity, social class], empowerment and values…..

7. PLOS authors have the option to publish the peer review history of their article (what does this mean?). If published, this will include your full peer review and any attached files.

Reviewer #1: **Yes: **Michelle Nakphong

Reviewer #2: **Yes: **Saraswathi Vedam

---

## [Author Response · Author response to Decision Letter 1]

12 Mar 2021

PONE-D-20-18688R1

Correlation among experience of person-centered maternity care, provision of care and women’s satisfaction: cross sectional study in Colombo, Sri Lanka

PLOS ONE

Reviewers' comments:

Reviewer's Responses to Questions

Comments to the Author

1. If the authors have adequately addressed your comments raised in a previous round of review and you feel that this manuscript is now acceptable for publication, you may indicate that here to bypass the “Comments to the Author” section, enter your conflict of interest statement in the “Confidential to Editor” section, and submit your "Accept" recommendation.

Reviewer #1: All comments have been addressed

Reviewer #2: (No Response)

2. Is the manuscript technically sound, and do the data support the conclusions?

Reviewer #1: Yes

Reviewer #2: Yes

3. Has the statistical analysis been performed appropriately and rigorously? 

Reviewer #1: Yes

Reviewer #2: Yes

4. Have the authors made all data underlying the findings in their manuscript fully available?

Reviewer #1: Yes

Reviewer #2: Yes

5. Is the manuscript presented in an intelligible fashion and written in standard English?

Reviewer #1: Yes

Reviewer #2: Yes

6. Review Comments to the Author

Reviewer #1: 

1) Thank you for your revision. The revised analyses and substantial discussion have greatly improved the paper. I especially appreciated the more detailed discussion surrounding women’s perceptions and normalization of disrespectful treatment. 

 *** Many thanks you for your appreciation 

2) You addressed some limitations with satisfaction as a measure, but there are other biases and limitations that should be addressed regarding satisfaction. For example, using satisfaction as a dichotomous variable, splitting satisfaction at 6 and above, will pose problems especially because respondents will often select the mid-point option as a cognitive bias. It appears that this is bias was also present in your sample, since over 15% of your sample selected 5. This suggests that a large fraction of those who selected the middle option were subject to cognitive biases and not influenced by other risk factors and likely overestimates the proportion of those who were “not satisfied.”

Rodway, P., Schepman, A., and Lambert, J. (2012). Preferring the One in the Middle: Further Evidence for the Centre-stage Effect. Applied Cognitive Psychology, 26 (2), 215-222 DOI: 10.1002/acp.1812

**Thank you for the interesting reference provided. Satisfaction was collected on a liker scale (1 to 10). However, when looking at the distribution of the satisfaction scores, Women’s overall satisfaction was not normally distributed (Shapiro-Wilk p<0.001) as shown in Fig1d. The median satisfaction score was 7 (IQR range: 5 to 9) with 295 women (73.7%) above the minimum satisfaction limit of 6; 186 (46.5%) had a satisfaction score between 6-8, and 109 (27.3%) a satisfaction score of >8 out of 10. We believe that this distribution does not suggest a cognitive bias. In addition, data on satisfaction had a good correlation with the PCMC scores (Spearman r= 0.58, p<0.001) (Fig 4C), with mothers who had a bad experience of PCMC expressing low satisfaction, and mothers with a good experience of PCMC reporting higher satisfaction, and thus suggesting that mothers genuinely expressed their satisfaction, based on their experience of care. 

3) Your discussion could also benefit by addressing limitations of the Bologna score. I am not sure that it can be considered a measure of “the provision of care” as defined by the WHO QOC framework for maternal and newborn health. Perhaps it might be better described as “recommended clinical practices” or something that highlights clinical practices. Both companion of choice and delivery in the position of choice are considered to be respectful care practices and I believe are classified under the “experience of care.” Because of this, I do not think the Bologna score can be considered an appropriate measure of the “provision of care.”

Additionally, it should be mentioned that the correlation between PCMC and Bologna care should be interpreted conservatively, since it may be partially due to overlapping items. For example, the PCMC scale has items relating to companionship during labor and delivery, and whether the woman was able to deliver in the position she wanted.

*** Many thanks for your thoughtful comment. The Bologna score include 5 indicators: 1) presence of a companion at the time of birth; 2) use of partograph; 3) absence of labor stimulation (use of oxytocin, external pressure of the uterine fundus, or episiotomy); 4) delivery in non-supine position; 5) skin-to-skin contact with their newborn immediately post-partum. You are very much right in saying that the first of this indicator is considered by the WHO framework as indicator of “experience of care”. However, all the other four are categorized under “provision of care”. In addition, labor companion and non-supine position at birth are known to also have an impact on outcomes such as the incidence of operative delivery, length of labor etc. For this reason, we generalized the Bologna score as a tool to assess primarily provision of care. We have made this clear in the paper now. 

We also would like to add that indictors of QOC is a complex construct which involves many overlapping parameters other than evidence-based practices alone. Generally, tools that are used to measure provision of care do not guarantee that they measure the provision of care alone, and many aspects of QOC have both component. 

We reported in the paper that the correlation in between Bologna and PCMC was very low (Pearson r=0.20, p<0.001). As you rightly pointed out, the PCMC include, out of 30 total items, the two items: “Were you allowed to have someone you wanted (outside of staff at the facility, such as family or friends) to stay with you during labour?” and “During the delivery, do you feel like you were able to be in the position of your choice? “These two items indeed refer to two items of the Bologna Score, but they explore the point from the angle of the experience of the mothers (ie. Were you allowed? do you feel like you were able ‘). This is a very different perspective/angle/lens from the one explored in the Bologna, which is merely factual. 

4) Minor points:

- Please address misspelled terms in footnotes of Table 1: “caesarian” “nera” “lapartotomy” *** This has been corrected 

- Line 304: “One third (28.5%)…” It seems that ‘over one-quarter’ or ‘nearly one-third’ would be more accurate than ‘one-third.’ *** This has been corrected

- Line 454: “heart” is misspelled *** This has been corrected

- The figures are quite blurry. I am not sure if this because the images are of low resolution or if the system has distorted images. *** This is due to the low resolution requested in the submission. We will provide higfh resolution images once the paper has been accepted

- Figure 3, please correct “Skin-Skin” *** This has been corrected

- Figure 4, especially because some of your graphs include plots using only discrete values (i.e. 4b), please consider illustrating density in your plots (for example, using “jitter” or dodging points). *** Ilaria 

Reviewer #2: 

1) Thank you for the opportunity to read this revised version. The paper is greatly improved in clarity and impact as a result of the edits and additions made in response to reviewers’ comments. 

*** Many thanks you for your appreciation

2) My remaining suggestions are mostly very minor copyediting recommendations; however, I do have one important note to the authors and editor: This study adds to the growing body of literature that shows mistreatment and human rights violations during childbearing is a global and widespread phenomena (1/6), and is even more prevalent among historically disenfranchised communities (2/3). It is an important contribution to our understanding not only frequency of disrespect and abuse, but that these person-centred metrics of quality care are important outcome in their own right (eg. They do not need to be linked to other maternal newborn adverse outcomes, or patient satisfaction to merit urgent attention). Yet, the authors appear to be hesitant to let the data speak to this mandate, reserving the emphasis in the discussion solely to the key metric they chose to use: patient satisfaction. Perhaps they could at least state more clearly in their conclusions (abstract and paper) that “findings indicate evidence of poor quality care across several domains of mistreatment in childbirth in Sri Lanka. [and] Patient satisfaction as an indicator of quality care is inadequate to inform health systems reform”. I can understand a reluctance to speak of prevalence of the most adverse outcomes (mistreatment) based on a sample size of 400 in the region (authors frame this analysis as pilot data), and I agree that discussions generally focus on findings per the originally intended methods. However, when results show egregious harm, in any study, it seems that researchers are obligated to name and elevate these findings and the urgency of rectifying harm and future targeted research on these matters. This is especially true when the findings align with incidence reported other published literature on the subject as cited.

*** Thank you for your comments. We sincerely appreciate this suggestion, and we have now revised the abstract and discussion.

Our findings indicate and reflect that the quality of care is poor across several domains of mistreatment during child birth in Sri Lanka. Patient satisfaction scores alone are inadequate to inform health systems reforms. 

Local authors have undertaken a series of focus group discussions to better examine women’s views (these data will be published in a separate publication). Based on data collected, a multifaceted intervention will be hopefully designed. 

3) Copy edits and typographical errors

Line 66 – Keywords - Suggested these edits to maximize searches for this article: Quality care, respectful maternity care, person-centered/person-centred, mistreatment, childbirth, disrespect and abuse (researchers in this field are unlikely to see satisfaction as an important component to include as noted by reviewer comments) *** This has been corrected

Line 96 (comma instead of full stop) *** This has been corrected

109-110 missing word: also defined [as] patient-centred, …*** This has been corrected

Line 142 suggest using “births” instead of deliveries – fast becoming the norm for respectful language (eg women give birth, providers *** This has been corrected

Line 293-296 The following sentence is unclear – either there were significant differences with some factors having lower scores or the scores were significantly lower for xxxx than xxxx*** This has been corrected

Additionally, other differences among the

294 full PCMC score (47.1; 95%CI 45.9-48.2) and “dignity and respect” (57.2; 95%CI 55.8-58.6), and 295 “supportive-care” subscores (50.5; 95%CI 49.0-51.9) were significantly lower (adjusted 296 p≤0.002 for all comparisons). PCMC not rescaled values are reported in Tables S2 and S3. *** This has been corrected

Line 303 awkward phrasing: 303 six (14.8%) felt to have been treated roughly – perhaps change to one out of six (14.8%) reported that they were treated roughly --or—one out of six (14.8%) reported mistreatment – all of the listed factors constitute mistreatment by global definitions. *** This has been corrected

Line 305 – best not to start a sentence with a percentage – perhaps rephrase: Most women (85.8%) reported that….*** This has been corrected

Line 357 women “at” their second pregnancy should be in their second pregnancy women….. or women [ in] their second pregnancy*** This has been corrected

422 mis-treated should be mistreated *** This has been corrected

4) Other comments

Line 368 “No factor was associated with women’s satisfaction” (sensitivity analysis just parity and non-supine position) -- this result shows that satisfaction is a poor discriminator of quality care and that finding should be discussed – with recommendation that future studies report on PCMC scores, and incidence of mistreatment as a stand alone indicator of quality, safety, rights. 

*** Thank you very much for your inspiring comment. We have added this in the discussion. 

5) Line in 394-399 that discusses above is too mild given the findings that 1/6 to ½ experience serious mistreatment – as written the discussion does not address that only one modifiable experience of care factor (birth position) was even moderately correlated – hence satisfaction is not measuring quality or the correlation shou 

*** Thank you very much for your inspiring comment. We have added this in the discussion. 

6) Lines 472-476 should immediately follow Line 427 as finding on differential experience by ethnicity are also socio-demographic predisposing factors. Then “ethnicity” should be added to the line 431 …-and better document how education, [ethnicity, social class], empowerment and values…..

 ***Very good suggestion, we have revised the text accordingly

---

## [Editor Report · Decision Letter 2]

16 Mar 2021

Correlation among experience of person-centered maternity care, provision of care and women’s satisfaction: cross sectional study in Colombo, Sri Lanka

PONE-D-20-18688R2

Dear Dr. Rishard,

We’re pleased to inform you that your manuscript has been judged scientifically suitable for publication and will be formally accepted for publication once it meets all outstanding technical requirements.

Kind regards,

Tanya Doherty, PhD

Academic Editor

PLOS ONE
---

## [Editor Report · Acceptance letter]

30 Mar 2021

PONE-D-20-18688R2 

Correlation among experience of person-centered maternity care, provision of care and women’s satisfaction: cross sectional study in Colombo, Sri Lanka 

Dear Dr. Rishard:

I'm pleased to inform you that your manuscript has been deemed suitable for publication in PLOS ONE. Congratulations! Your manuscript is now with our production department. 

Kind regards, 

on behalf of

Professor Tanya Doherty 

Academic Editor

PLOS ONE